# Finding good policies in average-reward Markov Decision Processes without prior knowledge

**Adrienne Tuynman**
adrienne.tuynman@inria.fr

**Rémy Degenne**
remy.degenne@inria.fr

**Emilie Kaufmann**
emilie.kaufmann@univ-lille.fr

Univ. Lille, Inria, CNRS, Centrale Lille, UMR 9189-CRIStAL, F-59000 Lille, France

## Abstract

We revisit the identification of an $\varepsilon$-optimal policy in average-reward Markov Decision Processes (MDP). In such MDPs, two measures of complexity have appeared in the literature: the diameter, $D$, and the optimal bias span, $H$, which satisfy $H \leq D$. Prior work have studied the complexity of $\varepsilon$-optimal policy identification only when a generative model is available. In this case, it is known that there exists an MDP with $D \simeq H$ for which the sample complexity to output an $\varepsilon$-optimal policy is $\Omega(SAD/\varepsilon^2)$ where $S$ and $A$ are the sizes of the state and action spaces. Recently, an algorithm with a sample complexity of order $SAH/\varepsilon^2$ has been proposed, but it requires the knowledge of $H$. We first show that the sample complexity required to estimate $H$ is not bounded by any function of $S$, $A$ and $H$, ruling out the possibility to easily make the previous algorithm agnostic to $H$. By relying instead on a diameter estimation procedure, we propose the first algorithm for $(\varepsilon, \delta)$-PAC policy identification that does not need any form of prior knowledge on the MDP. Its sample complexity scales in $SAD/\varepsilon^2$ in the regime of small $\varepsilon$, which is near-optimal. In the online setting, our first contribution is a lower bound which implies that a sample complexity polynomial in $H$ cannot be achieved in this setting. Then, we propose an online algorithm with a sample complexity in $SAD^2/\varepsilon^2$, as well as a novel approach based on a data-dependent stopping rule that we believe is promising to further reduce this bound.

## 1 Introduction

Reinforcement learning (RL) is a paradigm in which an agent interacts with its environment, modeled as a Markov Decision Process (MDP), by taking actions and observing rewards. Its goal is to learn, or to act according to, a good policy, that is a mapping from state to actions which maximizes cumulative rewards. However, there are different ways to define this notion of (expected) cumulative reward: in the *finite horizon* setting, one should maximize the expected sum of rewards up to a certain fixed horizon; in the *discounted* setting, each consecutive reward is $\gamma$ times as important as the previous one, with $0 < \gamma < 1$. In this paper, we focus on the *average reward* setting, in which the value of a policy is measured by its asymptotic mean reward per time step. This setting is ideal for long-term learning, as there is no need to tune the horizon or discount parameters. However, the asymptotic nature of its optimality criterion makes the problem more complicated and highly sensitive to small changes in the MDP, which is less observable in the finite horizon or discounted settings.

Formally, a Markov Decision Processes is defined as a tuple $(\mathcal{S}, \mathcal{A}, P, r)$ where $\mathcal{S}$ is the state space of finite size $S$, $\mathcal{A}$ is the action space of finite size $A$. Letting $\Sigma_\mathcal{X}$ denote the set of distribution

over a set $\mathcal{X}$, $P : \mathcal{S} \times \mathcal{A} \to \Sigma_\mathcal{S}$ is the (assumed unknown) transition kernel, and $r : \mathcal{S} \times \mathcal{A} \to \Sigma_{[0,1]}$ is the reward kernel. For each state-action pair $(s,a)$, we denote by $\bar{r}_{s,a} = \mathbb{E}[r(s,a)]$ the (assumed known) mean reward[1]. At each time step $t$, the agent observes a state $s_t \in \mathcal{S}$, takes an action $a_t$, and observes a reward $r_t \sim r(s_t, a_t)$ and a next state $s_{t+1} \sim P(s_t, a_t)$. In an average-reward MDP (AR-MDP) the value of a policy $\pi : \mathcal{S} \mapsto \Delta(\mathcal{A})$ is measured with its gain $g_\pi(s) = \lim_{T \to +\infty} \frac{1}{T} \mathbb{E}_\pi \left[ \sum_{t=0}^{T-1} r_t | s_0 = s \right]$ where the expectation is taken when the agent follows policy $\pi$ (i.e., $s_t \sim \pi(a_t)$) from the initial state $s$. In this paper, we consider weakly communicating MDPs (defined in Section 2) in which a policy $\pi_\star$ maximizing the gain in all states exists and has a constant value, denoted by $g_{\pi_\star}$.

We are interested in the best policy identification problem: we want to build an algorithm that learns the MDP by taking actions and collecting observations, until it can, after some (possibly random) number of interactions $\tau$ output a policy $\widehat{\pi}$ that is near-optimal. More precisely, given two parameters $\varepsilon \in (0,1)$ and $\delta \in (0,1)$ we seek an $(\varepsilon, \delta)$-Probably Approximately Correct (PAC) algorithm, that is an algorithm that satisfies

$$\mathbb{P}\left( \tau < \infty, \exists s \in \mathcal{S}, g_{\pi_\star} - g_{\widehat{\pi}}(s) > \varepsilon \right) \leq \delta .$$

Our objective is to find such an algorithm that has minimal sample complexity, i.e., that requires the least amount of steps $\tau$, with high probability or in expectation. Two different models can be considered for the collection of observations. In the online model, the algorithm can only choose the action $a_t$ to sample at each time step, as the state $s_t$ is determined by the MDP's dynamics and the previous actions. With a generative model however the agent can sample the reward and next state of any state action pair $(s_t, a_t)$, regardless of which state it arrived in at the previous time step.

To the best of our knowledge, prior work on $(\varepsilon, \delta)$-PAC best policy identification in AR-MDPs has exclusively considered the generative model setting. In the online setting, the regret minimization objective has however been studied a lot (e.g., [9, 2, 3]). Existing sample complexity or regret bounds feature different notions of complexity of the MDP, besides its size $S, A$. The diameter $D$ and the optimal bias span $H$, formally defined in the next section, are two such complexity notions that both feature in lower or upper bounds on the sample complexity of existing algorithms. More specifically for $(\varepsilon, \delta)$-PAC policy identification, the work of [22] provides a worse-case lower bound showing that there exists an MDP on which any $(\varepsilon, \delta)$-PAC algorithm using a generative model should have a sample complexity larger than $\Omega((SAD/\varepsilon^2)\log(1/\delta))$. The recent work of [27] provides an $(\varepsilon, \delta)$-PAC algorithm that takes $H$ as input and whose sample complexity is $\widetilde{\mathcal{O}}\left( (SAH/\varepsilon^2) \log(1/\delta) \right)$[2]. Using that $H \leq D$, this algorithm is thus optimal, and the lower bound is also in $\widetilde{\Omega}((SAH/\varepsilon^2)\log(1/\delta))$. This raises the following question: can an algorithm attain the same optimal sample complexity without prior knowledge of $H$? More broadly, as detailed in the next section, all existing $(\varepsilon, \delta)$-PAC algorithms require some form of prior knowledge on the MDP, and we propose the first algorithms that are agnostic to the MDP.

**Contributions** In the generative model setting, a first hope to get an algorithm agnostic to $H$ is to plug-in a tight upper bound on this quantity in the algorithm of [27]. Our first contribution is a negative result: the number of samples necessary to estimate $H$ within a prescribed accuracy is not polynomial in $H$, $S$ and $A$. This result is proved in Section 3. On the positive side, by combining a procedure for estimating the diameter $D$ inspired by [21] with the algorithm of [27] we propose in Section 4 Diameter Free Exploration (DFE), an algorithm for communicating MDPs that does not require any prior knowledge on the MDP and has a near-optimal $\widetilde{\mathcal{O}}\left( (SAD/\varepsilon^2) \log(1/\delta) \right)$ sample complexity in the asymptotic regime of small $\varepsilon$. Then in Section 5 we discuss the hardness of $(\varepsilon, \delta)$-PAC policy identification in the online setting. We notably prove a lower bound showing that the sample complexity of $(\varepsilon, \delta)$-PAC policy identification cannot always be polynomial in $S$, $A$ and $H$, even with the knowledge of $H$. On the algorithmic side, we propose in Section 6 an online variant of DFE whose sample complexity scales in $SAD^2/\varepsilon^2$. As prior work, DFE hinges on a conversion between the discounted and average-reward settings [22] and uses uniform sampling. Departing from this approach, we further propose a novel data-dependent stopping rule tailored for AR-MDPs that

---

[1] In most practical cases, the reward of the system are decided preemptively, and the uncertainty solely resides in the dynamics; moreover, estimating rewards is often easier than estimating the transition probabilities, and doing it can be done at little cost, therefore not changing our results much.

[2] In the paper, the $\widetilde{\mathcal{O}}$ hides constant factors and logarithmic terms in $S,A,D$ or $H$, $1/\varepsilon$ and $\log(1/\delta)$.

can be used both in the generative model and the online setting. We prove that it is $(\varepsilon, \delta)$-PAC for any (possibly adaptive) sampling rule and give preliminary sample complexity guarantees.

## 2 Related work

In order to position our work in the literature on best policy identification in average-reward MDP, we start by recalling some properties of average-reward MDPs and give a formal definition of the different complexity measures that have appeared in the literature.

First of all, it can be more or less easy to travel through the MDP and visit some states. We call an MDP weakly communicating if there exists a closed set of states that are all reachable from every other state (meaning $\mathcal{S}' \subset \mathcal{S}$ such that for any $s, s'$ in $\mathcal{S}'$, $\min_{\pi:\mathcal{S}\to\mathcal{A}} \mathbb{E}[\min\{t, s_t = s'\}|s_0 = s, \forall t', a_{t'} = \pi(s_{t'})] < +\infty$) and a possibly empty set of states which are transient (meaning the return time $\min\{t, s_t = s\}$ when the chain starts with $s_0 = s$ is not almost surely finite) under any policy. Furthermore, when there is no such transient state, the MDP is then communicating, and it is possible to define the diameter of the MDP to quantify how fast circulating in the MDP can be:

$$ D = \max_{s \neq s'} \min_{\pi:\mathcal{S}\to\mathcal{A}} \mathbb{E}[\min\{t > 0, s_t = s'\}|s_0 = s, \forall t', a_{t'} = \pi(s_{t'})] \,. $$

Finally, if the MDP satisfies for any policy $\pi$ and states $s, s'$ that $\mathbb{E}[\min\{t, s_t = s'\}|s_0 = s, \forall t', a_{t'} = \pi(s_{t'})] < +\infty$, then the MDP is ergodic. If the MDP satisfies this except for a possibly empty set of transient states, the MDP is then called unichain. In the following, the MDPs are all considered at least weakly communicating, though some results shall require some stronger assumptions, which we will specify. Notably, every mention of the diameter will assume communicating MDPs.

The diameter has appeared in previous upper and lower bound on the regret [9] and in the lower bound of [22] for best policy identification. The diameter is also used in the literature on the Stochastic Shortest Path (SSP) problem, in which one seek to minimize the sum of cost obtained before some goal state is reached (some details on this alternative framework are given in Appendix B). The SSP-diameter is defined as the maximum over states of the minimum expected steps to the goal state, and it appears in the sample complexity bounds given by [21]. We will see in the following that there are multiple similarities between the SSP setting and our average-reward setting, the use of the diameter being only one of them.

For a policy $\pi : \mathcal{S} \to \mathcal{A}$, we define the gain $g_\pi(s) = \lim_{T\to+\infty} \frac{1}{T} \mathbb{E}\left[\sum_{t=0}^{T-1} r_t | s_0 = s\right]$. By writing $P_\pi = (P(s, \pi(s)))_s$, and introducing $\overline{P_\pi} = \lim_{T\to+\infty} \frac{1}{T} \sum_{t=1}^{T} P_\pi^{t-1}$ the asymptotic distribution matrix, as well as defining $\overline{r}_\pi(s) = \mathbb{E}[r(s, \pi(s))]$, we have $g_\pi(s) = \overline{P_\pi}(s)\overline{r}_\pi$. Finally, we can define the bias vector $b_\pi = \sum_{t=1}^{+\infty} \left(P_\pi^{t-1} - \overline{P_\pi}\right) \cdot \overline{r}_\pi$. Those vectors satisfy the so-called Poisson equations (see [18]):

$$ \begin{cases} (I - P_\pi)g_\pi = 0 & (3) \\ g_\pi + (I - P_\pi)b_\pi = \overline{r}_\pi & (4) \end{cases} $$

We notice that the solution, $b_\pi$, is defined up to an additive element in $\mathrm{Ker}(I - P_\pi)$. We can then define $g^\star$ the optimal gain and (up to an additive constant) $b^\star$ the optimal bias satisfying

$$ g^\star + b^\star(s) = \max_a \left\{\overline{r}(s, a) + p_{s,a} b^\star\right\}, \quad (5) $$

where $p_{s,a}$ is the (row) vector of transition probabilities from $(s, a)$. We further define

$$ H = \max_s b^\star(s) - \min_s b^\star(s) $$

as the span of the optimal bias. The optimal bias span is always smaller than the diameter [2], which motivates a line of work replacing $D$ with $H$ in existing regret bounds [2, 7]. These improved bounds are however obtained only when $H$ is known. Similarly existing sample complexity bounds featuring $H$ all require this knowledge, as discussed in more detail below.

In addition to the diameter $D$ and the optimal bias span $H$, we define the mixing time

$$ t^\star_{\mathrm{mix}} = \max_\pi \inf\left\{t \geq 1 : \max_s \left\|P_\pi^t(s) - \overline{P_\pi}(s)\right\|_1 \leq \frac{1}{2}\right\} $$

Table 1: Existing sample complexity bounds in the generative model setting

| Algorithm | Bound | Setting |
|---|---|---|
| [23] | $\widetilde{\mathcal{O}}\left(\frac{SA(\tau t_{\mathrm{mix}}^{\star})^2}{\varepsilon^2}\log(1/\delta)\right)$ | $\frac{1}{\sqrt{\tau}S} \leq \overline{P_\pi}(s) \leq \frac{\sqrt{\tau}}{S}, \tau$ known $t_{\mathrm{mix}}^{\star}$ over all policies, known |
| [11] | $\widetilde{\mathcal{O}}\left(\frac{SA(t_{\mathrm{mix}}^{\star})^2}{\varepsilon^2}\log(1/\delta)\right)$ | $t_{\mathrm{mix}}^{\star}$, known |
| [12] | $\widetilde{\mathcal{O}}\left(\frac{SAt_{\mathrm{mix,D}}^{\star}}{\varepsilon^3}\log(1/\delta)\right)$ | $t_{\mathrm{mix,D}}^{\star}$, known |
| [16] | $\widetilde{\mathcal{O}}\left(\frac{SA(t_{\mathrm{mix}}^{\star})^3}{\varepsilon^2}\log(1/\delta)\right)$ | $t_{\mathrm{mix}}^{\star}$, known (policy gradient) |
| [24] | $\widetilde{\mathcal{O}}\left(\frac{SAt_{\mathrm{mix,D}}^{\star}}{\varepsilon^2}\log(1/\delta)\right)$ | $t_{\mathrm{mix,D}}^{\star}$, known |
| [22] | $\widetilde{\mathcal{O}}\left(\frac{SAH}{\varepsilon^3}\log(1/\delta)\right)$ | $H$ known |
| [26] | $\widetilde{\mathcal{O}}\left(\left[\frac{SAH^2}{\varepsilon^2} + \frac{S^2AH}{\varepsilon}\right]\log(1/\delta)\right)$ | $H$ known |
| [27] | $\widetilde{\mathcal{O}}\left(\frac{SAH}{\varepsilon^2}\right)$ | $H$ known |
| This paper | $\widetilde{\mathcal{O}}\left(\left[\frac{SAD}{\varepsilon^2} + S^2AD^2\right]\log(1/\delta)\right)$ | no prior knowledge |
| [12] | $\Omega\left(\frac{SAt_{\mathrm{mix}}^{\star}}{\varepsilon^2}\right)$ | Lower bound (worse case) |
| [22] | $\Omega\left(\frac{SAD}{\varepsilon^2}\log(1/\delta)\right)$ | Lower bound (worse case) |

which has also appeared in previous sample complexity bounds. By default the maximum is taken over all stochastic policies $\pi : \mathcal{S} \rightarrow \Sigma_{\mathcal{A}}$, and we denote by $t_{\mathrm{mix,D}}^{\star}$ the same quantity where the maximum is restricted to deterministic policies.

Multiple papers have considered the problem of finding an $\varepsilon$-optimal policy in average-reward with access to a generative model. We summarize existing results in Table 1. [12] and [22] derived worse case lower bounds for the problem by exhibiting MDPs on which a certain number of samples *must* be collected to guarantee correctness of any $(\varepsilon, \delta)$-PAC algorithm. The literature has first focused on the mixing time as a measure of the complexity of MDPs [23, 11, 12, 16], until the mixing time-scaling lower-bound for the sample complexity was matched by [24], for an algorithm taking $t_{\mathrm{mix},D}^{\star}$ as input. Algorithms using the optimal bias span $H$ were introduced more recently, by [22] and [26], and the lower bound scaling in $H$ was matched up to logarithmic factors by [27]. This side of the literature has mostly used the links between discounted and average-reward MDPs and has used that for known $H$ it is possible to choose a discount $\gamma$ (with $1 - \gamma$ of order $\varepsilon/H$) such that $H$-optimal policies in the discounted MDP are $\varepsilon$-optimal in the average-reward MDP [22]. This idea leads to upper bounds of the sample complexity that scale with the (assumed known) upper bound on $H$. To make these algorithms agnostic to the MDP, a natural question is therefore: can we find a tight upper bound on $H$ from the data?

## 3   On the hardness of estimating $H$

In this section, we investigate the complexity of estimating the optimal bias span $H$ and more specifically of finding upper bounds on this quantity.

**Definition 1.** *We say an algorithm computes a $\Delta$-tight upper bound for the optimal bias span with probability $1 - \delta$ when, on any MDP of optimal bias $H$, it outputs $\hat{H}$ such that with probability higher than $1 - \delta$, $H \leq \hat{H} \leq H + \Delta$.*

Theorem 1 shows that there exists an MDP on which the sample complexity of an algorithm finding a $\Delta$-tight upper bound with probability larger than $1 - \delta$ can be arbitrarily large. As a consequence, there cannot exist a bound on this sample complexity depending solely on $S$, $A$, $H$, $\delta$ and $\Delta$.

**Theorem 1.** *For any $\delta < \frac{1}{2e^4}$, $T > 0$, $\Delta$, there exists an ergodic MDP $\mathcal{M}$ with optimal bias span $H = 1/2$, $S = 3$ and $A = 2$ such that any algorithm that computes a $\Delta$-tight upper bound for the optimal bias span with probability $1 - \delta$ under a generative model assumption needs (in expectation) more than $T$ samples in $\mathcal{M}$.*

**Sketch of proof**   To give an idea of the proof, we first focus on the weakly-communicating setting, with the full proof and the necessary ergodicity transformation detailed in Appendix A. We use the family of MDPs $(\mathcal{M}_R)_{0 < R < 1}$ displayed in Figure 1. Indeed, having $R < 1/2$ makes the full-line policy optimal; since moving from one state to the optimal final state (1) is very easy in this case, no state is drastically worse than another, hence a small optimal bias span, $H = 1/2$. However, $R > 1/2$ makes the dashed-line policy optimal; moving from state 3 to state 2 is more difficult as $p$ is small, taking on average $1/p$ time steps, and state 3 has a worse reward than state 2; thus state 3 is considered way worse than state 2, hence a high optimal bias span, $H = \frac{1+p}{p}R$. Thus, to bound $H$ tightly, one needs to know whether $R > 1/2$ or $R < 1/2$, which can require many samples if $R$ is very close to $1/2$.

$\square$

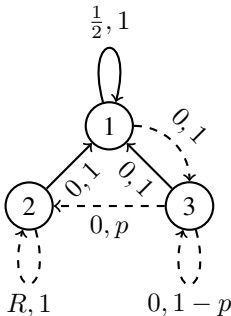

Figure 1: MDP $\mathcal{M}_R$, the hard instance for Theorem 1. Each arrow corresponds to a state-action and next state combination, and is annotated with the mean reward of the action and the probability of the transition. Arrows with a different line style correspond to different actions.

We remark that since the hard MDP instance in our proof has a fixed optimal bias span $1/2$, we would have the same problem if we looked for algorithms trying to find $\hat{H}$ such that, with high probability, $H \leq \hat{H} \leq (1 + \Delta)H$. Moreover, instead of looking for upper bound we could also consider the task of estimating $H$ within a given margin of error and the same issue would arise.

While Theorem 1 does not preclude the existence of an algorithm that is agnostic to $H$ and reaches a $\widetilde{\mathcal{O}}((SAH/\varepsilon^2) \log(1/\delta))$ sample complexity, it still suggests that assuming a known upper bound on $H$ is a lot to ask. In the next section we will see that for communicating MDPs such an assumption is not needed to attain a near-optimal sample complexity.

## 4   A near-optimal algorithm without prior knowledge

As we have seen, finding tight upper bounds on $H$ is not feasible in finite time. In the literature on regret minimization, in which improved regret bounds featuring $H$ have also been derived when $H$ is known, two types of workarounds have been used. In the REGAL algorithm[3], [2] propose to use a doubling trick to counter not knowing $H$, at the expense of an additional factor of $\sqrt{S}$ in the regret. The doubling trick method is not applicable to BPI problems, though. For communicating MDPs, the idea of plugging-in an upper bound on $D$ which is also a (not necessarily tight) upper bound on $H$ was first proposed by [25] and permits to match the minimax lower bound on the regret of [9] featuring $D$. In this section, we translate this idea to the best policy identification setting.

We propose Diameter Free Exploration (DFE), stated as Algorithm 1, which combines a diameter estimation sub-routine that follows from the work of [21, 20] with the state-of-the-art algorithm of

---

[3]Some issues about this algorithm are detailed in [6], but the doubling trick still deserves consideration.

**Data:** Accuracy $\varepsilon \in (0,1)$, confidence level $\delta \in (0,1)$

Let $\widehat{D}$ be the output of Algorithm 2 with accuracy 1 and confidence level $\delta/2$.

Let $\widehat{\pi}$ be the output of Algorithm 3 with accuracy $\varepsilon$, upper bound $\overline{H} = \widehat{D}$ and confidence level $\delta/2$

**return** $\hat{\pi}$

**Algorithm 1:** Diameter Free Exploration (DFE)

[27] when (an upper bound on) $H$ is known. Both components are described in details in Appendix B with their theoretical properties. More precisely, DFE first calls Algorithm 2 to output a quantity $\widehat{D}$ which satisfies $D \leq \widehat{D} \leq 4D$ with probability larger than $1 - \delta/2$, using a random number of samples $N$ from each state action pair $(s,a)$ that satisfies $N = \widetilde{\mathcal{O}}(D^2(\log(1/\delta) + S))$. Then Algorithm 3 (which also uses uniform sampling) takes as input $\widehat{D}$ and is guaranteed to output a policy that is $\varepsilon$-optimal with probability larger than $1 - \delta/2$, using a total number of samples of order $\widetilde{\mathcal{O}}\left(\frac{SA\widehat{D}}{\varepsilon^2} \log\left(\frac{1}{\delta}\right)\right)$. This gives the following theorem.

**Theorem 2.** *Algorithm 1 is $(\varepsilon, \delta)$-PAC and its sample complexity satisfies, with probability $1 - \delta$,*

$$\tau = \widetilde{\mathcal{O}}\left(\left[\frac{SAD}{\varepsilon^2} + D^2 SA\right] \log\left(\frac{1}{\delta}\right) + D^2 S^2 A\right) .$$

From Theorem 4 of [22], there exists a communicating MDP with a sample complexity in $\Omega\left(\frac{SAD}{\varepsilon^2} \ln(1/\delta)\right)$. Therefore, the DFE algorithm is worse than the lower bound by a multiplicative factor $\left(1 + D\varepsilon^2 + \frac{DS\varepsilon^2}{\log 1/\delta}\right)$. In particular, in the regime of small $\varepsilon$, this algorithm is optimal up to logarithmic factors.

## 5 On the hardness of best policy identification in the online setting

In the online setting, many algorithms with regret guarantees have been designed, however to the best of our knowledge there exists no online algorithm that is guaranteed to output a policy $\widehat{\pi}$ satisfying $g_{\widehat{\pi}} \geq g^* - \varepsilon$ with probability larger than $1 - \delta$[4]. In this section, we give some elements of explanation.

**On the hardness of a regret to PAC conversion** In the finite-horizon setting, [10] gave a way to convert any algorithm guaranteeing sublinear regret into a BPI algorithm with finite sample complexity. However, in average-reward MDP, this conversion is not straightforward. Indeed, it hinges on the fact that selecting at random one of the policies played during the regret-minimizing algorithm should yield a good enough policy: the more time steps there are, the smaller the regret induced by the most recent policy is, thus the better the policy is. However, in our setting, we know that the empirical mean reward of a policy converges towards the gain of this policy, but the speed of this convergence is determined by the mixing time. Moreover, as shown in [22], this mixing time can be arbitrarily large. However, the regret is not *defined* asymptotically. In [9], the regret at time $T$ is defined as $Tg^* - \mathbb{E}[\sum_{t=1}^{T} r_t]$. In [17], the regret is $\mathbb{E}_{\pi^*}[\sum_{t=1}^{T} r_t] - \mathbb{E}[\sum_{t=1}^{T} r_t]$. As the mixing time can be big, it is possible that these regret measures are small for a large number of steps, but that the underlying policy is not anywhere near asymptotically optimal.

It is therefore not possible to preemptively define the number of steps necessary for the regret-minimizing algorithm to find an asymptotically good policy, and not just one that is good on the short term. For example, for any $N \in \mathbb{N}$, $N \geq 2$, consider the MDP $\mathcal{M}_p$ displayed in Figure 2 with $p = \frac{1}{N}$ and any $p' < \frac{1-\varepsilon}{1+\varepsilon} p$. While the dashed-line policy is asymptotically best, the full-lined policy is better for at least $N$ time steps, and is not even $\varepsilon$-optimal, as we show in Appendix C.1. An algorithm guaranteeing low regret (over a slightly modified version of the MDP to guarantee ergodicity) could therefore take the full-line action for a number of steps independent of the known parameters $S$ and $A$, and the knowledge of $N$ would be required to know how many samples are

---

[4]The work of [16] provides an online policy gradient algorithm for ergodic MDPs but its guarantees are not expressed in the PAC setting considered in the paper (rather with simple regret). Moreover, this algorithm requires some prior knowledge on the MDP through some parameter $\beta$ related to the mixing time

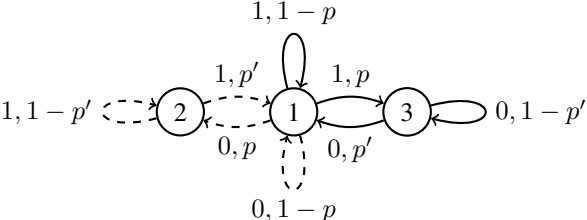

Figure 2: MDP $\mathcal{M}_{p,p'}$ with high mixing time.

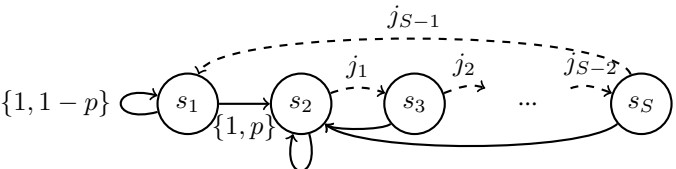

Figure 3: MDP $\mathcal{M}_j$, the hard instance for Theorem 3

necessary for best policy identification. This means that a wrapper turning any regret minimization algorithm into a best policy identification algorithm would require estimating and incorporating this measure $N$ somewhere, which argues against the existence of a straightforward wrapper as in the episodic setting. Similar arguments in the SSP setting have been brought up for example in [21].

**A hardness result**  As the following lower bound shows, it is in fact not possible at all to have an online best policy identification algorithm that has a sample complexity bound polynomial in $S$, $A$ and $H$. This result is as a counterpart of the hardness result proved by [5] in SSP-MDPs.

**Theorem 3.** *For any $S \geq 4$, $A \geq 4$, $\varepsilon \in \left(0, \frac{1}{4}\right)$, $\delta \in \left(0, \frac{1}{16}\right)$, and any $(\varepsilon, \delta)$-PAC online algorithm, there exists a weakly communicating MDP with $S$ states, $A$ actions, mean rewards in $[0, 1]$, $H \leq S$ and $D \geq A^{S-1}/16\varepsilon$ such that if the algorithm starts in a given state $s_1$, $\mathbb{E}[\tau] = \Omega\left(\frac{A^{S-1}}{\varepsilon}\right)$.*

**Sketch of proof**  For $j \in (S-1)^A$, we define the hard instance $\mathcal{M}_j$ as in Figure 3. An agent in $s_2$ must execute the precise sequence of actions $j$ to get back to $s_1$, which is the only state that generates non zero rewards. Since not learning this precise sequence would mean the agent will perform badly on one of the $\mathcal{M}_j$, and therefore perform a very suboptimal policy, the agent will need to have a big enough sample size to at least see $s_2$ with high probability when starting in $s_1$. With a small value of $p$, it is possible to make the sample complexity exponentially big, as we detail in Appendix C.2.

This hard MDP is actually inspired from the hard SSP-MDP instance considered by [5] to prove that sample-efficient learning is impossible in the online setting of SSP. Indeed, their hard MDP is one where the cost is always the same in each state, in which case we can easily transform the SSP problem into an average reward one, as shown in Figure 5. Indeed, if the costs of the SSP-MDP are all equal and non-zero, then finding an optimal policy in it can be show to be equivalent to finding an optimal policy in the corresponding AR-MDP in which the only reward is in the initial state. □

We remark that while the hard MDP instances $\mathcal{M}_j$ used in our proof have an optimal bias span $H$ that is upper bounded by $S$, its diameter is exponential is $S$. Hence while Theorem 3 implies that no online algorithm can get a $\mathcal{O}\left(\frac{SAH}{\varepsilon^2}\right)$ sample complexity (with or without the knowledge of $H$) on every instance and for every $\varepsilon$, it does not rule out the possibility to have an online algorithm with a sample complexity scaling with $D$. We propose such an algorithm in the next section.

# 6 Algorithms for the online setting

## 6.1 Diameter Free Exploration for the online setting

Our first idea to tackle the online setting is to propose an online variant of the Diameter Free Exploration algorithm from Section 4. While the first ingredient of DFE is Algorithm 2, a diameter estimation procedure based on a generative model, we could use instead the online diameter estimation procedure proposed by [20] to get an upper bound $D \leq \widehat{D} \leq 4D$ with probability larger than $1 - \delta$ using a slightly worse sample complexity in $\bar{\mathcal{O}}_\delta(D^3 SA)$[5]. The second ingredient, Algorithm 3, starts by collecting a fixed number of samples $\overline{n}$ proportional to $(\widehat{D}/\varepsilon^2) \log(12SA/(\delta\varepsilon))$ from each $(s,a)$, using the generative model. We can replace this step by using the GOSPRL algorithm of [20] to collect this amount of samples with an online algorithm, which requires $\bar{\mathcal{O}}_\delta(SAD\overline{n} + D^{3/2}S^2 A)$ samples, with high probability. We call the resulting algorithm Online-DFE.

**Theorem 4.** *Online-DFE is $(\varepsilon, \delta)$-PAC and its sample complexity satisfies, with probability larger than $1 - \delta$, $\tau = \bar{\mathcal{O}}_\delta \left( \frac{SAD^2}{\varepsilon^2} + S^2 AD^3 \right)$.*

Compared to Theorem 2 for the generative model setting, we note a multiplicative $D$ factor in the sample complexity, as well as a less explicit dependency in $\delta$. Still, it provides an online PAC algorithm whose sample complexity scales in $SAD^2/\varepsilon^2$ in the small $\varepsilon$ regime. We remark that the extra factor $D$ compared to the lower bound for the generative setting comes from the use of GOSPRL and the (mostly theoretical) conversion from the generative to the online setting. To get more efficient algorithms, a promising direction is to investigate more adaptive algorithms, directly tailored for the average-reward setting. This is what we do next.

## 6.2 Towards more adaptive algorithms

We recall that an identification algorithm uses three components: a sampling rule, selecting at time $t$ a state-action pair $(s_t, a_t)$, after which the next state $s_{t+1} \sim P(s_t, a_t)$ and the reward $r_t \sim r(s_t, a_t)$ is observed; a stopping rule $\tau$ which decides whether the data collection can be stopped, and a recommendation $\widehat{\pi}$ which is a guess for a near-optimal policy that depends on the first $\tau$ observations. In the online setting, the sampling rule is actually reduced to the choice of $a_t$ given $s_t$, as we have the constraint that $s_{t+1} \sim P(s_t, a_t)$. In this section we introduce a novel stopping rule, that can be used in conjunction with different sampling rule, possibly in the online setting.

**The Value Iteration stopping rule** We first introduce some notation to define the stopping rule. We denote by $N^n_{s,a} = \sum_{t=1}^{n} \mathbb{1}((s_t, a_t) = (s, a))$ the number of visits to the state action pair $(s, a)$ in the first $n$ steps and $\hat{p}^n_{s,a}$ the empirical estimate of the transition probability vector $p_{s,a}$ built at step $n$, where $p^n_{s,a}$ can be arbitrarily chosen to be the constant vector equal to $\frac{1}{S}$ if $N^n_{s,a} = 0$.

Our stopping rule takes inspiration from the stopping condition of value iteration for average-reward MDPs. It relies on a sequence of bias vectors $b_n \in \mathbb{R}^S$ and on the ability to build confidence intervals on the quantity $I_{s,a}(n) = \overline{r}_{s,a} + p_{s,a}b_n - b_n(s)$. This leads us to build an upper bound of the span of $I_{s,a}(n)$, which can be seen as the average-reward equivalent of the Bellman error as defined in [8]. We know that, if $b_n = b^\star$, then $(\max_a I_{s,a}(n))_s$ is equal to the constant vector $g^\star$. Conversely, by Theorem 8.5.5 of [18], if the span of $(\max_a I_{s,a}(n))_s$ is small, then the associated policy $\pi(s) = \arg\max_a I_{s,a}(n)$ is approaching the optimal one; this can be used to derive a criterion for correctness of value iteration in average-reward MDPs. Therefore, by examining an upper-bound on the span of $I$, we get the following theorem.

**Theorem 5.** *In a weakly communicating MDP, for a given vector $b_n \in \mathbb{R}^S$, if for all pairs $(s, a)$ we have $p_{s,a}b_n \in \left( L^n_{s,a}(b_n; \delta), U^n_{s,a}(b_n; \delta) \right)$, then defining $\hat{\pi}_n(s) = \arg\max_a I^{n,\flat}_{s,a}(b_n; \delta)$, we have*

$$\min_s \max_a I^{n,\flat}_{s,a}(b_n; \delta) \leq g_{\hat{\pi}_n} \leq g^\star \leq \max_s \max_a I^{n,\sharp}_{s,a}(b_n; \delta)$$

*where $I^{n,\flat}_{s,a}(b_n; \delta) := \overline{r}_{s,a} + L^n_{s,a}(b_n; \delta) - b_n(s)$ and $I^{n,\sharp}_{s,a}(b_n; \delta) := \overline{r}_{s,a} + U^n_{s,a}(b_n; \delta) - b_n(s)$.*

---

[5] the $\bar{\mathcal{O}}_\delta$ notation hides constants and logarithmic terms in $S,A,D,H$ and $1/\delta$ (instead of $\log(1/\delta)$ in $\widetilde{\mathcal{O}}$) as the work of [20] did not try to optimize the dependency in $\delta$, as we did in Section 4.

Now to propose a concrete stopping rule, we provide expressions of $U$ and $L$ chosen such that the premise of Theorem 5 is satisfied in every round $n$, with high probability. $\mathrm{KL}(p,q)$ denotes the Kullback-Leibler divergence between the probability vectors $p$ and $q$ on $\mathcal{S}$.

**Definition 2.** *Given a sequence of $(b_n)_n$, we define*

$$
U_{s,a}^n(b_n;\delta) = \max\left\{p'b_n \middle| N_{s,a}^n \mathrm{KL}(\hat{p}_{s,a}^n, p') \leq x(\delta, N_{s,a}^n)\right\}
$$

$$
L_{s,a}^n(b_n,\delta) = \min\left\{p'b_n \middle| N_{s,a}^n \mathrm{KL}(\hat{p}_{s,a}^n, p') \leq x(\delta, N_{s,a}^n)\right\}
$$

*for the threshold function $x(\delta, y) = \log(SA/\delta) + (S-1)\log(e(1+y/(S-1)))$ and let*

$$
\tau_{\varepsilon,\delta} = \min_n\left\{\max_s\max_a I_{s,a}^{n,\sharp}(b_n,\delta) - \min_s\max_a I_{s,a}^{n,\flat}(b_n;\delta) \leq \varepsilon\right\} . \tag{6}
$$

The following result is a consequence of Theorem 5 and a KL-based time uniform concentration result for the transition probabilities from [1] (Lemma 5 in Appendix). Using Pinsker's inequality (see Remark 1 in Appendix where we discuss some computational aspects), we can also justify that this theorem hold for $U_{s,a}^n$, $L_{s,a}^n$ replaced by

$$
\widetilde{U}_{s,a}^n(b_n;\delta) = \hat{p}_{s,a}^n b_n + ||b_n||_\infty\sqrt{\frac{2x(\delta, N_{s,a}^n)}{N_{s,a}^n}}, \quad \widetilde{L}_{s,a}^n(b_n;\delta) = \hat{p}_{s,a}^n b_n - ||b_n||_\infty\sqrt{\frac{2x(\delta, N_{s,a}^n)}{N_{s,a}^n}} . \tag{7}
$$

**Theorem 6.** *For any sampling rule $((s_n, a_n))_n$, and any sequence $(b_n)_n$ of bias vectors, the algorithm using the stopping rule $\tau = \tau_{\varepsilon,\delta}$ defined in (6) and recommending $\widehat{\pi}(s) = \arg\max_a I_{s,a}^{\tau,\flat}(b_\tau;\delta)$ satisfies $\mathbb{P}(\tau < \infty, g^\star - g_{\widehat{\pi}} > \varepsilon) \leq \delta$.*

Theorem 6 shows that for any sampling rule, be it in the generative model or in the online setting, the stopping rule (6) and associated recommendation rule yields an $(\varepsilon,\delta)$-PAC algorithm, for any choice of bias vector sequence $b_n$. Of course, bad choices of sampling rules and bias vectors could still yield $\tau = \infty$ almost surely (e.g. picking always $(s_t, a_t) = (s_1, a_1)$ when a generative model is available).

**Choosing a sampling rule** As a sanity-check we analyze in Appendix D.2 an algorithm which combines the simplest possible sampling rule, uniform sampling from a generative model, with the VI stopping rule (6) where the sequence of biais vector is $b_n = \hat{b}_n$ where $\hat{b}_n$ is the optimal bias vector in the AR-MDP with rewards $\bar{r}_{s,a}$ and transition probabilities given by $(\hat{p}_{s,a}^n)_{s,a}$ (see Algorithm 4). We prove in Theorem 9 that in unichain MDPs, for sufficiently small $\varepsilon$ the sample complexity of this algorithm is bounded with probability larger than $1 - \delta$ by $\widetilde{\mathcal{O}}\left(\frac{SA(H\vee\Gamma_\mathcal{M})^2}{\varepsilon^2}\left(\log\frac{1}{\delta} + S\right)\right)$, where $\Gamma_\mathcal{M}$ is some constant depending on the MDP. We refer the reader to Appendix D.2 for the definition of this constant, that has a complex expression and should be in most case larger than $H$.

We remark that we could also use GOSPRL to turn Algorithm 4 into an online algorithm (using phases of increasing length in which we collect a uniform number of samples using GOSPRL and check our stopping rule), with a sample complexity essentially multiplied by $D$. However the resulting sample complexity is likely to be much larger than $SAD^2/\varepsilon^2$ in the small $\varepsilon$ regime, making this algorithm not very interesting. We believe that to get efficient algorithms for the online setting it is important to depart from this uniform sampling + GOSPRL approach. Our stopping rule actually suggests some clever online choices that could be used to make the algorithm stop earlier. For example it could be interesting to make a greedy choice of the bias vector $b_n$ that minimizes over $b \in \mathbb{R}^S$ the quantity $\max_s\max_a I_{s,a}^{n,\sharp}(b;\delta) - \min_s\max_a I_{s,a}^{n,\flat}(b;\delta)$. Assuming we could compute this vector, a possible online sampling rule could be optimistic choice $a_n = \mathrm{argmax}_a[\bar{r}_{s,a} + U_{s_n,a}^n(b_n,\delta)]$. We leave the analysis of such algorithms for future work.

## 7 Conclusion and perspective

We provided several elements indicating that the optimal bias span $H$ may not be an adequate complexity measure for $\varepsilon$-optimal policy identification in an average-reward MDP. In the generative model setting, as all existing algorithms with sample complexity featuring $H$ require prior knowledge

of this quantity, we investigated the question of estimating $H$ and proved that the sample complexity of this estimation task can be arbitrarily large. Then, in the online setting, we gave a lower bound on the sample complexity indicating that no algorithm can get a $SAH/\varepsilon^2$ sample complexity on every MDP, with or without the knowledge of $H$. On the algorithmic side, we proposed the first best policy identification algorithms for the generative model setting that does not require any form of prior knowledge on the MDP. By estimating the diameter $D$ instead of $H$, DFE attains a near-optimal sample complexity in $SAD/\varepsilon^2$ for communicating MDPs. We further proposed an online variant of DFE with a slightly larger $SAD^2/\varepsilon^2$ sample complexity, which is a factor $D$ away from the worse case lower bound established in the generative model setting. We leave as an open question whether there exists online algorithms with a $SAD/\varepsilon^2$ sample complexity. In future work, we will investigate whether the novel adaptive stopping rule proposed in our work can lead to this reduced sample complexity, when combined with a suitable online sampling rule.

## Acknowledgments and Disclosure of Funding

The authors acknowledge the funding of the French National Research Agency under the project FATE (ANR22-CE23-0016-01) and the PEPR IA FOUNDRY project (ANR-23-PEIA-0003). The authors are members of the Inria team Scool.

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

# A  Proof of Theorem 1

We only prove here the result for weakly communicating MDPs, and provide the ergodic MDP necessary to prove the general theorem.

Fix $\delta$, $T$ and $\Delta$. Define $d = e^4$ and $d' = 128$. Let $\varepsilon = \min\left\{\frac{1}{10}, \frac{\Delta}{2}, \frac{1}{2\sqrt{\frac{32Td'}{-\ln 2d\delta}+1}}\right\}$, $p = \frac{\frac{1}{2}+\varepsilon}{\Delta-\varepsilon}$. To ease the notation, we denote by $\mathcal{M}_\varepsilon$ and $\mathcal{M}'_\varepsilon$ respectively the MDPs $\mathcal{M}_{\frac{1}{2}-\varepsilon}$ and $\mathcal{M}_{\frac{1}{2}+\varepsilon}$ in Figure 1. The rewards distribution in these MDPs are assumed to be Bernoulli variables with the means indicated in the figure.

The optimal policy in $\mathcal{M}_\varepsilon$ always chooses the actions represented by a full line. Its gain is $\frac{1}{2}$, its bias vector (up to an additive constant) is $(0, -\frac{1}{2}, -\frac{1}{2})$. In $\mathcal{M}'_\varepsilon$, the optimal policy always chooses the dashed actions and its optimal gain is $\frac{1}{2} + \varepsilon$. Using the Poisson equations, we can show that its associated bias vector (up to an additive constant) is $\left(-\left(\frac{1}{2}+\varepsilon\right)\frac{1+p}{p}, 0, -\left(\frac{1}{2}+\varepsilon\right)\frac{1}{p}\right)$. With $H_\varepsilon$ the span of the optimal bias for $\mathcal{M}_\varepsilon$ and $H'_\varepsilon$ that of $\mathcal{M}'_\varepsilon$, we have:

$$
\begin{aligned}
H'_\varepsilon - H_\varepsilon &= \frac{1}{p}\left(\left(\frac{1}{2}+\varepsilon\right)(1+p) - \frac{p}{2}\right) = \frac{1}{p}\left(\frac{1}{2}+\varepsilon\right) + \varepsilon \\
&= \frac{\Delta-\varepsilon}{\frac{1}{2}+\varepsilon}\left(\frac{1}{2}+\varepsilon\right) + \varepsilon \\
&= \Delta
\end{aligned}
$$

Consider now an algorithm that outputs $\hat{H}$ a $\Delta$-tight upper bound for the optimal bias span with probability greater than $1-\delta$ on any MDP. We denote by $P$ and $P'$ the probability with regards to $\mathcal{M}_\varepsilon$ and $\mathcal{M}'_\varepsilon$ respectively, and $\mathbb{E}$ and $\mathbb{E}'$ the associated expectation.

We denote by $\tau$ the total number of samples used before stopping, by $\hat{T}$ the number of samples of the dashed action taken in state 2 before stopping and by $K_t$ the number of times the agent gets a reward of 1 among the first $t$ visits of state 2. We introduce the three events

$$
\begin{aligned}
\mathcal{E}_1 &= \{\hat{H} < H'_\varepsilon\} \\
\mathcal{E}_2 &= \left\{\max_{1\leq t\leq t^\star} \left|\left(\frac{1}{2}-\varepsilon\right)t - K_t\right| \leq z\right\} \\
\mathcal{E}_3 &= \{\hat{T} \leq t^\star\},
\end{aligned}
$$

where we define $z = \sqrt{2\left(\frac{1}{2}-\varepsilon\right)\left(\frac{1}{2}+\varepsilon\right)t^\star\ln\frac{d}{\theta}}$, with $\theta = \exp\left(\frac{-4d'\varepsilon^2 t^\star}{\left(\frac{1}{2}-\varepsilon\right)\left(\frac{1}{2}+\varepsilon\right)}\right)$ and let $t^\star = \frac{\left(\frac{1}{2}-\varepsilon\right)\left(\frac{1}{2}+\varepsilon\right)}{4d'\varepsilon^2}\ln\frac{1}{2d\delta}$. We let $\mathcal{E} = \mathcal{E}_1 \cap \mathcal{E}_2 \cap \mathcal{E}_3$.

Let $W$ be the interaction history of the learner and the generative model, $L(w) = P(W = w)$ and $L'(w) = P'(W = w)$. For $K = K_{\hat{T}}$,

$$
\frac{L'(W)}{L(W)}\mathbb{1}_\mathcal{E} = \frac{(1/2+\varepsilon)^K(1/2-\varepsilon)^{\hat{T}-K}}{(1/2-\varepsilon)^K(1/2+\varepsilon)^{\hat{T}-K}}\mathbb{1}_\mathcal{E} \geq \frac{\theta}{d}\mathbb{1}_\mathcal{E}
$$

by the arguments within the proof for Lemma 13 of [5]. Using a change of measure, we can write

$$
P'(\mathcal{E}_1) \geq P'(\mathcal{E}) = \mathbb{E}'[\mathbb{1}_\mathcal{E}(W)] = \mathbb{E}\left[\frac{L'(W)}{L(W)}\mathbb{1}_\mathcal{E}(W)\right] \geq \frac{\theta}{d}P(\mathcal{E}) = 2\delta P(\mathcal{E}). \tag{8}
$$

Moreover, in $\mathcal{M}_\varepsilon$, $K_t - (1/2 - \varepsilon)t$ is the sum of $t$ Bernoulli variables of expectation $(1/2 - \varepsilon)$, and thus of variance $(1/2 - \varepsilon)(1/2 + \varepsilon)$. Kolmogorov's inequality states that

$$
P(\overline{\mathcal{E}_2}) = P\left(\max_{1\leq t\leq t^\star}\left|K_t - \left(\frac{1}{2}-\varepsilon\right)t\right| > \varepsilon\right) \leq \frac{t^\star\left(\frac{1}{2}-\varepsilon\right)\left(\frac{1}{2}+\varepsilon\right)}{\varepsilon^2}
$$

which entails $P(\mathcal{E}_2) \geq 7/8$. The detailed computation is given in the proof for Lemma 12 of [5].

Now, we assume towards contradiction that $P(\mathcal{E}_3) \geq 7/8$. Using the above, this yields

$$
\begin{aligned}
P\left(\overline{\mathcal{E}}\right) &\leq P(\overline{\mathcal{E}_1}) + P(\overline{\mathcal{E}_2}) + P(\overline{\mathcal{E}_3}) \\
&\leq \delta + \frac{2}{8} \\
&< \frac{1}{2},
\end{aligned}
$$

where the first inequality uses that $P(\mathcal{E}_1) \geq 1 - \delta$ by the correctness of the algorithm and the second inequality uses that $\delta \leq \frac{1}{2e^4}$. Hence, we have $P(\mathcal{E}) > \frac{1}{2}$ which leads to $P'(\mathcal{E}_1) > \delta$ using (8). This contradicts the correctness of the algorithm. Therefore, we have $P(\mathcal{E}_3) < \frac{7}{8}$.

It follows that with probability larger than $\frac{1}{8}$, $\hat{T} \geq t^\star$ on $\mathcal{M}_\varepsilon$, hence

$$
\mathbb{E}[\tau] \geq \mathbb{E}[\hat{T}] \geq \frac{t^\star}{8} \geq T,
$$

which concludes the proof.

More generally, adding the transitions described in Figure 4 with $\tau' = \frac{\tau p}{1+p}$ can be used to show that computing a $\frac{\Delta}{1+\tau}$-tight upper bound for the optimal bias span with high probability would need in expectation more than $T$ samples.

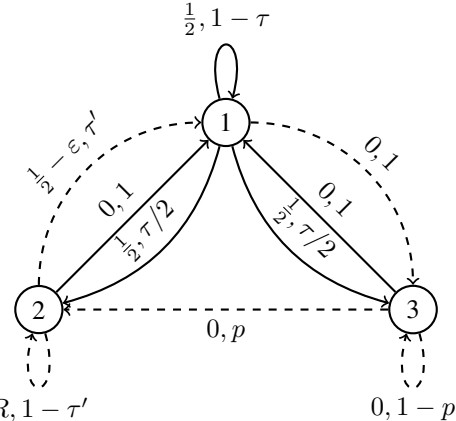

Figure 4: $\widetilde{\mathcal{M}}_R$ in the ergodic case

# B  Complements for Section 4

In this Appendix, we provide some details on the two components of our near-optimal algorithm: a procedure to provide a high-probability upper bound on the diameter (Algorithm 2) and a near-optimal algorithm for best policy identification in an average reward MDP that requires an upper bound on $H$ as an input (Algorithm 3). Finally, we give the proof of Theorem 2.

## B.1  Diameter Estimation Procedure

The procedure proposed by [20] for estimating the diameter hinges on the observation that the diameter can expressed in terms of optimal values in a goal-oriented Markov Decision Decision (or SSP-MDP for Stochastic Shortest Path). We recall that in a SSP-MDP the transition kernel is coupled with a goal state $s_g \in \mathcal{S}$ and a cost function $c : (s, a) \mapsto c(s, a)$. The value of a (deterministic) policy $\pi$, that is to be minimized, is denoted by

$$
V_{c,s_g}^\pi(s) = \mathbb{E}^\pi \left[ \sum_{t=1}^{\tau_\pi^{(s_g)}(s)} c(s_t, \pi(s_t)) \middle| s_1 = s \right]
$$

where $\tau_\pi^{(s_g)}(s) = \inf\{t : s_{t+1} = s_g | s_1 = s, \pi\}$ is the number of steps before reaching the goal when starting from state $s$ and following policy $\pi$. The diameter can thus be written

$$\begin{aligned} D &= \max_s \max_{s' \neq s} \min_\pi \mathbb{E}[\tau_\pi^{(s)}(s')] = \max_s \max_{s' \neq s} \min_\pi V_{\mathbf{1},s}^\pi(s') \\ &= \max_s \max_{s' \neq s} V_{\mathbf{1},s}^\star(s') \end{aligned}$$

where $V_{c,s_g}^\star$ denotes the optimal (i.e. minimal) value function in the SSP-MDP with cost $c$ and goal state $s_g$ and $\mathbf{1}$ is the cost function that is constant and equal to 1.

Prior works on SSP-MDPs [19, 21] have proposed and analyzed an Extended Value Iteration scheme for SSP, whose goal is to obtain confidence bounds on $V_{c,s_g}^\star$, that we now recall for completeness. Given a cost function $c$ and a goal state $s_g$, EVI-SSP$_{c,s_g}$ takes as input a set of plausible transitions $\mathcal{P} = (\mathcal{P}(s,a))_{s,a}$ such that $\mathcal{P}(s,a)$ are included in the set of probability distribution over $\mathcal{S}$ (and the unknown vectors $p_{s,a}$ is believed to belong to $\mathcal{P}(s,a)$), and a precision $\mu_{\text{VI}}$. It introduces the extended optimal Bellman operator, defined for $v \in \mathbb{R}^S$ as

$$\widetilde{L}v(s) = \min_{a \in \mathcal{A}} \left[ c(s,a) + \min_{\widetilde{p} \in \mathcal{P}(s,a)} \widetilde{p}v \right]$$

for all $s \neq s_g$ and $\widetilde{L}v(g) = 0$. EVI-SSP$_{c,s_g}$ sets $v_0 = \mathbf{0} \in \mathbb{R}^S$ and defines $v_{j+1} = \widetilde{L}v_j$ for $j \geq 0$. It stops at iteration $\widetilde{j} = \min\{j : \|v_{j+1} - v_j\| \leq \mu_{\text{VI}}\}$ and outputs $\widetilde{v} = v_{\widetilde{j}}$. It also outputs an optimistic transition kernel $\widetilde{p} = (\widetilde{p}_{s,a})_{s,a}$, defined for all $s, a$ as

$$\widetilde{p}_{s,a} \in \underset{\widetilde{p} \in \mathcal{P}(s,a)}{\operatorname{argmin}} \ \widetilde{p}v$$

and the optimistic greedy policy

$$\widetilde{\pi}(s) = \underset{a \in \mathcal{A}}{\operatorname{argmin}} \ [c(s,a) + \widetilde{p}v] \ .$$

We denote by $\widetilde{V}_{c,s_g}^\pi(s)$ the value function of policy $\pi$ in the SSP with cost function $c$, goal state $s_g$ and transition kernel $\widetilde{p}$. Previous work (see e.g. Lemma 4 of [21]) have established the following properties.

**Lemma 1.** *Assume that, for all $(s,a)$, $p_{s,a} \in \mathcal{P}(s,a)$. Letting $(\widetilde{v}, \widetilde{p}, \widetilde{\pi}) = \text{EVI-SSP}_{c,s_g}(\mathcal{P}, \mu_{VI})$, the following inequalities hold (component-wise):*

*(i)* $\widetilde{v} \leq V_{c,s_g}^\star$ *and* $\widetilde{v} \leq \widetilde{V}_{c,s_g}^\star \leq \widetilde{V}_{c,s_g}^{\widetilde{\pi}}$

*(ii)* *if* $\mu_{VI} \leq \frac{c_{\min}}{2}$ *then* $\widetilde{V}_{c,s_g}^{\widetilde{\pi}} \leq \left(1 + \frac{2\mu_{VI}}{c_{\min}}\right) \widetilde{v}.$

Notably, this result gives an upper bound on the optimal value in the optimistic SSP-MDP. With the additional help of a simulation lemma, proved by [20], we can further relate it to the value in the true SSP-MDP.

**Lemma 2** (Lemma 2 in [21]). [6] *Let $p$ and $\widetilde{p}$ be two transition kernels such that for all $(s,a)$, $\|p_{s,a} - \widetilde{p}_{s,a}\|_1 \leq \eta$. Assume that $c_{\min} > 0$ and that under both $p$ and $\widetilde{p}$ there exists at least one policy that reaches $s_g$ almost surely from any state (such a policy is called proper). Let $\pi$ be a proper policy in $\widetilde{p}$ such that*

$$2\eta \|\widetilde{V}_{c,s_g}^\pi\|_\infty \leq c_{\min}. \tag{9}$$

*Then $\pi$ is proper in $p$ and*

$$\forall s \in \mathcal{S}, V_{c,s_g}^\pi(s) \leq \left(1 + \frac{2\eta \|\widetilde{V}_{c,s_g}^\pi\|_\infty}{c_{\min}}\right) \widetilde{V}_{c,s_g}^\pi(s)$$

---

[6]Compared to the original statement, we fixed a small typo in the condition (9), and also propagated the changes accordingly in the definition of Algorithm 2 and its analysis.

Finally, the last step to be able to apply Lemma 1 is to construct the sets $\mathcal{P}(s,a)$ that contains the true transition probabilities with high probability. This can be done as in the rest of the paper by relying on Lemma 5, which gives confidence regions of the form

$$\{p \in \Sigma_S : \|p - \hat{p}_{s,a,n}\|_1 \leq B(n,\delta)\}$$

where $\hat{p}_{s,a,n}$ is the empirical estimate of the transition probability after the $n$-th transition has been observed from $(s,a)$ and $B(n,\delta) = \sqrt{\frac{2\log(SA/\delta) + 2(S-1)\log(e(1+n/(S-1)))}{n}}$.

The resulting algorithm for estimating the diameter is described in Algorithm 2. It is a slight variation of the procedures proposed by [20] to estimate the SSP diameter and by [21] to estimate the diameter in the online setting. In particular, our instantiation relies on simpler confidence regions.

---

**Data:** Accuracy $\varepsilon > 0$, confidence level $\delta \in (0,1)$
Set $W = \frac{1}{2}$ and $\widetilde{v}_\infty = 1$
**while** $\widetilde{v}_\infty > W$ **do**
  $\quad W \leftarrow 2W$
  $\quad$ Set accuracy $\eta = \frac{\varepsilon}{4W}$ and compute $N = N(\delta,\eta) = \inf\{n : B(n,\delta) \leq \eta\}$
  $\quad$ Call the generative model until each $(s,a)$ gets $N$ visits.
  $\quad$ Let each $(s,a)$ let $\mathcal{P}(s,a) = \{p \in \Sigma_S : \|p - \hat{p}_{s,a,N}\|_1 \leq \eta\}$ and $\mathcal{P} = (\mathcal{P}(s,a))_{s,a}$.
  $\quad$ **for** $s = s_1, ..., s_S$ **do**
  $\quad\quad$ Let $(\widetilde{v}^{(s)}, \widetilde{p}^{(s)}, \widetilde{\pi}^{(s)}) = \text{EVI-SSP}_{1,s}\left(\mathcal{P}, \frac{1 \wedge \varepsilon}{2}\right)$
  $\quad$ **end**
  $\quad$ Let $\widetilde{v}_\infty = \max_s \max_{s' \neq s} \widetilde{v}^{(s)}(s')$.
**end**
**return** $\widehat{D} = (1 + \eta\widetilde{v}_\infty/2)\widetilde{v}_\infty$

**Algorithm 2:** A diameter estimation procedure

---

**Theorem 7.** *Let $\varepsilon \leq 1$. With probability $1 - \delta$, Algorithm 2 run with parameters $\varepsilon$ and $\delta$ outputs an estimate $\widehat{D}$ which satisfies*

$$D \leq \widehat{D} \leq \left(1 + \frac{\varepsilon(1+\varepsilon)}{2}\right)(1+\varepsilon)D$$

*using $SA \times N(\delta, \frac{\varepsilon}{8D})$ samples where*

$$N(\delta,\eta) := \inf\{n > 0 : B(n,\delta) \leq \eta\} ,$$

*leading to an overall sample complexity of*

$$\widetilde{\mathcal{O}}\left(\frac{D^2 SA}{\varepsilon^2}\log\left(\frac{1}{\delta}\right) + \frac{D^2 S^2 A}{\varepsilon^2}\right)$$

*where the $\widetilde{\mathcal{O}}$ ignores logarithmic factors in $S, A, D, 1/\varepsilon$ and $\log(1/\delta)$.*

*Proof.* The proof closely follows that of [20]. We let $W_n$ be the value of $W$ in the $n$-th iteration of the while loop (starting at $n = 1$), such that $W_n = 2^{n-1}$. We define $\eta_n = \frac{\varepsilon}{4W_n}$ and $\widetilde{v}_n$ the value of $\widetilde{v}_\infty$ at the end of the $n$-th iteration. The number of iteration used by the algorithm is $\widetilde{n} = \min\{n : \widetilde{v}_n \leq W_n\}$.

Assume the event

$$\mathcal{E} = (\forall s, a, \forall n \geq 1, \|\hat{p}_{s,a,n} - p_{s,a}\|_1 \leq B(n,\delta))$$

holds. Using a property of the EVI scheme (first statement in Lemma 1), for all $n$, $\widetilde{v}_n^{(s)} \leq V_{1,s}^\star \leq D$ (component-wise), which yields $\widetilde{v}_n \leq D$. Hence $\widetilde{n}$ is bounded by $\min\{n : D \leq W_n\} \leq \log_2(D) + 1$. We now bound the final accuracy $\widetilde{\eta} := \eta_{\widetilde{n}}$. First, as $\widetilde{v}_\infty := \widetilde{v}_{\widetilde{n}} \leq W_{\widetilde{n}}$, we have $\widetilde{\eta} \leq \frac{\varepsilon}{4\widetilde{v}_\infty}$. Moreover, using that

$$D \geq \widetilde{v}_{\widetilde{n}-1} > W_{\widetilde{n}-1} = \frac{W_{\widetilde{n}}}{2}$$

further yields

$$\frac{\varepsilon}{8D} \leq \widetilde{\eta} \leq \frac{\varepsilon}{4\widetilde{v}_\infty}.$$

We let $\widetilde{v}^{(s)}, \widehat{p}^{(s)}, \widetilde{\pi}^{(s)}$ be the output of Extended Value Iteration with goal state $s$ in the final iteration, and let $\widetilde{V}_{\mathbf{1},s}$ denote the value function in the SSP MDP with costs $\mathbf{1}$, goal state $s$ and transition kernel $\widehat{p}^{(s)}$. For every $s, s'$, using the simulation lemma, we have

$$V^{\star}_{\mathbf{1},s}(s') \;\leq\; V^{\widetilde{\pi}^{(s)}}_{\mathbf{1},s}(s') \leq (1 + 2\widetilde{\eta}\|\widetilde{V}^{\widetilde{\pi}^{(s)}}_{\mathbf{1},s}\|_\infty)\widetilde{V}^{\widetilde{\pi}^{(s)}}_{\mathbf{1},s}(s')$$

provided that the condition

$$2\widetilde{\eta}\|\widetilde{V}^{\widetilde{\pi}^{(s)}}_{\mathbf{1},s}\|_\infty \leq 1$$

is satisfied. This is the case by using the second statement of Lemma 1 together with the upper bound on $\widetilde{\eta}$ established above:

$$2\widetilde{\eta}\|\widetilde{V}^{\widetilde{\pi}^{(s)}}_{\mathbf{1},s}\|_\infty \leq 2\widetilde{\eta}(1+\varepsilon)\|\widetilde{v}^{(s)}\|_\infty \leq 2\widetilde{\eta}(1+\varepsilon)\widetilde{v}_\infty \leq \frac{\varepsilon(1+\varepsilon)}{2} \leq 1 \;,$$

where the last step uses that $\varepsilon \leq 1$. Hence, one can write

$$
\begin{aligned}
V^{\star}_{\mathbf{1},s}(s') \;&\leq\; (1 + 2\widetilde{\eta}\|\widetilde{V}^{\widetilde{\pi}^{(s)}}_{\mathbf{1},s}\|_\infty)\widetilde{V}^{\widetilde{\pi}^{(s)}}_{\mathbf{1},s}(s') \\[4pt]
&\overset{(a)}{\leq}\; (1 + 2\widetilde{\eta}(1+\varepsilon)\|\widetilde{v}^{(s)}\|_\infty)(1+\varepsilon)\widetilde{v}^{(s)}(s') \\[4pt]
&\overset{(b)}{\leq}\; (1 + 2\widetilde{\eta}(1+\varepsilon)\widetilde{v}_\infty)(1+\varepsilon)\widetilde{v}_\infty = \widehat{D} \\[4pt]
&\overset{(c)}{\leq}\; \left(1 + \frac{(1+\varepsilon)\varepsilon}{2}\right)(1+\varepsilon)D,
\end{aligned}
$$

where $(a)$ uses the second statement of Lemma 1, $(b)$ uses the definition of $\widetilde{v}_\infty$ and $(c)$ uses that $\widetilde{v}_\infty \leq D$ and $\widetilde{\eta}v_\infty \leq \varepsilon/4$. Recalling that $D = \max_{s,s'\neq s} V^{\star}_{\mathbf{1},s}(s')$ yields that, on event $\mathcal{E}$,

$$D \leq \widehat{D} \leq (1 + \frac{(1+\varepsilon)\varepsilon}{2})(1+\varepsilon)D.$$

Moreover, the number of samples collected per transition is $N(\delta, \widetilde{\eta}) \leq N\left(\delta, \frac{\varepsilon}{8D}\right)$.

The conclusion follows from the fact that $\mathcal{E}$ holds with probability larger than $1 - \delta$ (Lemma 5) and by upper bounding $N\left(\delta, \frac{\varepsilon}{8D}\right)$ using Lemma 6 in Appendix E. Specifically, choosing the parameters $\Delta^2 = \eta^2$, $a = 2\log\left(\frac{SA}{\delta}\right)$, $b = 2(S-1)$, $c = e$ and $d = \frac{e}{S-1}$ we get

$$N(\delta, \eta) \leq \frac{1}{\eta^2}\log\left(\frac{SA}{\delta}\right) + \frac{2(S-1)}{\eta^2}\log\left(e + \frac{e}{(S-1)\eta^4}\log\left(\frac{SA}{\delta}\right) + \frac{8e}{\eta^4}\right).$$

which leads to the approximation stated in Theorem 7. $\qquad\square$

## B.2 Near-optimal algorithm using the knowledge of $H$

We recall here for completeness the algorithm of [27] to find an $\varepsilon$-optimal policy in an average reward MDP when an upper bound $\overline{H}$ of the optimal bias span $H$ is known, and its theoretical guarantees. The algorithm consists in a slight variation of the Perturbed Empirical Model-Based Planning algorithm originally given by [15] (for which a refined analysis was proposed by [27]) with the reduction from the average reward to the discounted case from [22].

---

**Data:** Accuracy $\varepsilon \in (0, 1]$, upper bound $\overline{H}$ on $H$, confidence level $\delta \in (0, 1)$
  Set discount factor $\overline{\gamma} = 1 - \frac{\varepsilon}{12\overline{H}}$
Set $\overline{n} = \frac{144C_2\overline{H}}{\varepsilon^2}\log\left(\frac{12SA}{\delta\varepsilon}\right)$ with $C_2$ the constant in Theorem 1 of [27]
Collect $\overline{n}$ samples from the transition in each $(s, a)$
Let $\hat{p}$ be the estimated transition kernel based on all transitions collected
Compute the randomized reward function $\widetilde{r}(s, a) = r(s, a) + X_{s,a}$ where $X_{s,a} \overset{i.i.d}{\sim} \mathcal{U}\left(\left[0, \frac{\varepsilon}{72}\right]\right)$
Compute $\hat{\pi}$ the optimal policy in the discounted MDP $(\overline{\gamma}, \hat{p}, \widetilde{r})$
**return** $\hat{\pi}$

**Algorithm 3:** Algorithm 2 from [27]

---

**Theorem 8.** *With probability $1 - \delta$, Algorithm 3 with parameters $\varepsilon \in (0, 1]$, $\overline{H}$ satisfying $H \leq \overline{H}$ and $\delta \in (0, 1)$ outputs a policy $\hat{\pi}$ satisfying*

$$\mathbb{P}\left(\forall s \in \mathcal{S}, \rho^\star - \rho^{\hat{\pi}}(s) \leq \varepsilon\right) \geq 1 - \delta$$

*and collects a (deterministic) total number of transitions given by*

$$\frac{144 C_2 S A \overline{H}}{\varepsilon^2} \log\left(\frac{12 S A}{\delta \varepsilon}\right),$$

*where $C_2$ is the constant in Theorem 1 of [27].*

### B.3 Proof of Theorem 2

We let $\widehat{D}$ be the output of Algorithm 2 run with parameters 1 and $\delta/2$ and $\tau_1$ the total number of samples it uses. From Theorem 7, it holds that

$$\mathbb{P}\left(D \leq \widehat{D} \leq 4D, \tau_1 = \widetilde{\mathcal{O}}\left(D^2 S A \log(1/\delta) + D^2 S^2 A\right)\right) \geq 1 - \delta/2.$$

We let $\widehat{\pi}$ be the output of Algorithm 3 run with parameter $\varepsilon$, $\widehat{D}$ and $\delta/2$ and $\tau_2$ the total number of samples it uses. Using Theorem 8, we have

$$\mathbb{P}\left(g^\star - g_{\widehat{\pi}} \leq \varepsilon, \tau_2 \leq \frac{144 C_2 S A \widehat{D}}{\varepsilon^2} \log\left(\frac{24 S A}{\delta \varepsilon}\right) \,\middle|\, D \leq \widehat{D}\right) \geq 1 - \delta/2.$$

The total sample complexity being $\tau = \tau_1 + \tau_2$, using a union bound yields that with probability $1 - \delta$, it holds that $g^\star - g_{\widehat{\pi}} \leq \varepsilon$ and

$$
\begin{aligned}
\tau &\leq \widetilde{\mathcal{O}}\left(D^2 S A \log(1/\delta) + D^2 S^2 A\right) + \frac{576 C_2 S A D}{\varepsilon^2} \log\left(\frac{24 S A}{\delta \varepsilon}\right) \\
&= \widetilde{\mathcal{O}}\left(\left[\frac{S A D}{\varepsilon^2} + D^2 S A\right] \log\left(\frac{1}{\delta}\right) + D^2 S^2 A\right) \\
&= \widetilde{\mathcal{O}}\left(\left[\frac{S A D}{\varepsilon^2} + D^2 S^2 A\right] \log\left(\frac{1}{\delta}\right)\right).
\end{aligned}
$$

## C  Complements for Section 5

### C.1  Study of the MDP $\mathcal{M}_{p,p'}$

We recall the MDP introduced in Figure 2. We prove the following: for any $N$, for any $p < 1/N$ and any $p' < \frac{1-\varepsilon}{1+\varepsilon} p$, the policy in full line $\pi$ is optimal for more than $N$ steps, but is not $\varepsilon$-optimal in average reward.

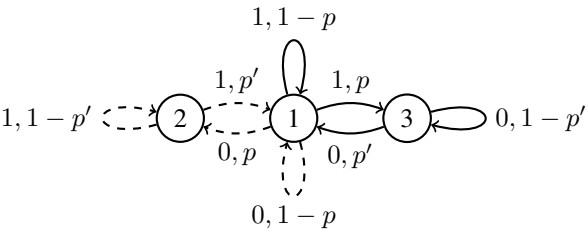

Figure 2: MDP $\mathcal{M}_{p,p'}$ (repeated from page 7)

For that, we define $u_n$ the probability of being in state 1 at timestep $n$. We note that $u_n$ does not depend on the policy chosen. Since we know $\begin{cases} u_0 = 1 \\ u_{n+1} = (1-p) u_n + p'(1 - u_n) \end{cases}$, we have

$$u_n = (1 - p - p')^n \left(1 - \frac{p'}{p + p'}\right) + \frac{p'}{p + p'}$$

Therefore, the (asymptotic) gain for the full-line policy $\pi$ is $g_\pi = \frac{p'}{p+p'}$, while that of the dashed-line policy $\pi'$ is $g_{\pi'} = 1 - \frac{p'}{p+p'} = \frac{p}{p+p'}$. Since $p' < \frac{1-\varepsilon}{1+\varepsilon}p$,

$$p'(1+\varepsilon) < p(1-\varepsilon)$$
$$p' < p - (p+p')\varepsilon$$
$$g_\pi < g_{\pi'} - \varepsilon$$

and we have indeed that $\pi$ is not $\varepsilon$-optimal.

Moreover, the empirical reward up to step $n$ for policy $\pi$ is $r_n = \sum_{t=0}^n u_n$, and $r'_n = n+1-\sum_{t=0}^n u_n$ for policy $\pi'$. As $(r_n/n)_n$ is decreasing and $(r'_n/n)_n$ is increasing, policy $\pi'$ becomes better than $\pi$ when $\sum_{t=0}^n u_n \le \frac{n+1}{2}$, that is, when

$$(n+1)\frac{p'}{p+p'} + \left(1 - \frac{p'}{p+p'}\right)\sum_{t=0}^n (1-p-p')^t \le \frac{n+1}{2}$$

$$\frac{p}{p+p'} \cdot \frac{1}{n+1} \cdot \frac{1-(1-p-p')^{n+1}}{p+p'} \le \frac{1}{2} - \frac{p'}{p+p'}$$

which implies

$$\frac{p}{p+p'} \cdot \frac{1}{n+1} \cdot \frac{(n+1)(p+p') - n(n+1)(p+p')^2}{p+p'} \le \frac{1}{2} - \frac{p'}{p+p'}$$

by using that $(1-x)^{n+1} < 1 - (n+1)x + n(n+1)x^2/2$ for any $0 < x < 1$. Finally, $\pi'$ being better than $\pi$ at timestep $n$ implies $\frac{-n}{2} \le \left(\frac{1}{2} - \frac{p'}{p+p'}\right)\frac{1}{p} - \frac{1}{p+p'}$, i.e., $n \ge \frac{1}{p} \ge N$.

## C.2 Proof of Theorem 3

*Proof.* For $S, A, \varepsilon, \delta$ satisfying the assumptions of Theorem 3, we let $p = \frac{16\varepsilon}{A^{S-1}}$ and define the family of MDPs $\mathcal{M}_j$ for any $j \in A^{S-1}$, displayed in Figure 3. There are $A$ actions available in each of the $S$ states, but the figure only displays actions with distinct transitions.

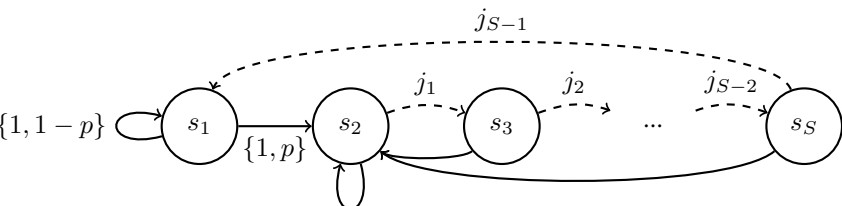

Figure 3: MDP $\mathcal{M}_j$ (repeated from page 7)

- In state $s_1$, for any action $a$, a reward of 1 is incurred, and there is probability $p$ of getting into state $s_2$, and probability $1-p$ of staying in $s_1$.

- In state $s_n$ for $n \in \{2, \dots, S\}$, taking action $j_{n-1}$ transitions to state $s_{n+1}$ (where $s_{S+1} = s_1$) and every other action transitions to state $s_2$, with no reward.

To summarize, it is very unlikely to get into state $s_2$; but, once it has been entered, only one specific deterministic policy can allow an agent to get back to the favorable state $s_1$.

Let $\pi$ be a (possibly stochastic) policy. Let us write $p_i^j = \pi(j_{i-1}|s_i)$ for $i \in \{2, \dots S\}$, and $x_\pi^j = \prod_{i=2}^S p_i^j$. With $P_{\pi,j}$ the transition matrix under $\pi$ in MDP $\mathcal{M}_j$, and $\overline{P}_{\pi,j} = \lim_{T\to\infty} \frac{1}{T}\sum_{t=1}^T P_{\pi,j}^{t-1}$, we know that $P_{\pi,j}\overline{P}_{\pi,j} = \overline{P}_{\pi,j}$. Since the MDP $\mathcal{M}_j$ is unichain, by theorem A.2 of [18], all the rows of $\overline{P}_{\pi,j}$ are identical to a vector $(\eta_1, \dots \eta_S) \in \Sigma_S$.

We therefore deduce the following linear system:

$$
\begin{cases}
\eta_1 = (1-p)\eta_1 + p_S^j \eta_S \\[2mm]
\eta_2 = p\eta_1 + \displaystyle\sum_{n=2}^{S}(1-p_n^j)\eta_n \\[2mm]
\forall n \in \{3,...S\}, \eta_n = p_{n-1}^j \eta_{n-1} \\[2mm]
\displaystyle\sum_{n=1}^{S}\eta_n = 1
\end{cases}
$$

Injecting the third equation into the second yields $\eta_2 = p\eta_1 + \sum_{n=2}^{S-1}(\eta_n - \eta_{n+1}) + \eta_S - x_\pi^j \eta_2$ and finally $x_\pi^j \eta_2 = p\eta_1$. Therefore,

$$
\eta_1\left(1 + \frac{p(1 + p_2^j + p_2^j p_3^j + ... + x_\pi^j)}{x_\pi^j}\right) = 1 \implies \eta_1 \leq \frac{1}{1 + \frac{p}{x_\pi^j}},
$$

using that $1 + p_1^j + p_2^j p_3^j + ... + x_\pi^j \geq 1$. Finally,

$$
g_\pi^j \leq \frac{1}{1 + \frac{p}{x_\pi^j}}.
$$

The optimal policy in $\mathcal{M}_j$ is to play action $j_{n-1}$ in state $s_n$ for $n \in \{2, \ldots, S\}$, so the above equation becomes $\eta_1(1 + p(S-1)) = 1$ and the optimal gain in $\mathcal{M}_j$ is $g^j = \frac{1}{1+p(S-1)}$.

If a policy $\pi$ is $\varepsilon$-optimal in $\mathcal{M}_j$, we have

$$
g_\pi^j \geq g^j - \varepsilon
$$
$$
\frac{1}{1 + \frac{p}{x_\pi^j}} \geq \frac{1}{1 + p(S-1)} - \varepsilon
$$
$$
(S-1)p \geq \frac{p}{x_\pi^j} - \varepsilon\left(1 + (S-1)p + \frac{p}{x_\pi^j} + \frac{(S-1)p^2}{x_\pi^j}\right)
$$
$$
x_\pi^j \geq p\frac{1 - \varepsilon(1 + (S-1)p)}{(S-1)p + \varepsilon(1 + (S-1)p)}
$$
$$
(S-1)x_\pi^j \geq \frac{1}{1 + \varepsilon\frac{((S-1)p+1)^2}{(S-1)p(1-\varepsilon-\varepsilon(S-1)p)}}
$$

Since $(S-1)p \leq \frac{1}{4}$, $(1 + (S-1)p)^2 \leq 2$. Since we also have $\varepsilon < \frac{1}{4}$, $(1 - \varepsilon - \varepsilon(S-1)p) \geq \frac{1}{2}$. Therefore, $(S-1)x_\pi^j \geq \frac{1}{1 + \frac{4\varepsilon}{(S-1)p}}$. Finally, $\frac{4\varepsilon}{(S-1)p} \geq 1$, and $x_\pi^j \geq \frac{p}{8\varepsilon}$. Since $\sum_j x_\pi^j = 1$, we deduce that each policy $\pi$ can be $\varepsilon$-optimal for at most $\frac{8\varepsilon}{p}$ MDPs in $\{\mathcal{M}_j\}_j$. By denoting $y_\pi^j = \mathbb{1}_{\{\pi \text{ is } \varepsilon\text{-optimal in } \mathcal{M}_j\}}$, we have $\sum_j y_\pi^j \leq \frac{8\varepsilon}{p}$.

For the remainder of the proof, we can use the proof of Theorem 9 of [5], which we retranscribe here. Assume towards contradiction that there exists a learner that is $(\varepsilon, \delta)$-correct whose sample complexity is smaller than $1/p$ with probability larger than $7/8$ on all the instances $(\mathcal{M}_j)_j$. By writing $\mathcal{E}_1$ the event that the first $1/p$ steps from $s_1$ all transit to $s_1$; $\mathcal{E}_2$ the event that the learner uses at most $1/p$ samples; and $\mathcal{E} = \mathcal{E}_1 \cap \mathcal{E}_2$; we can prove that for every $j$, writing $P_j$ the probability distribution w.r.t. $\mathcal{M}_j$, we have $P_j(\mathcal{E}) \geq \frac{1}{8}$.

We let $\mathcal{E}'$ be the bad event in which the policy $\hat\pi$ returned by the learner is not $\varepsilon$-optimal. We observe that on the event $\mathcal{E}$, the distribution of the output policy $\hat\pi$ conditionally to $\mathcal{E}$ is independent on $j$ as on the event $\mathcal{E}$, the algorithm has never visited another state than $s_1$ before stopping and outputting $\hat\pi$. We denote by $P(\hat\pi|\mathcal{E})$ the common value of $P_j(\hat\pi|\mathcal{E})$ for all $j$, which satisfies $\int_{\hat\pi} P(\hat\pi|\mathcal{E})d\hat\pi = 1$.

It follows that $\sum_j \int_{\hat{\pi}} P(\hat{\pi}|\mathcal{E}) y_{\hat{\pi}}^j d\hat{\pi} \leq \frac{8\varepsilon}{p}$ and that there exists $j$ such that $\int_{\hat{\pi}} P_j(\hat{\pi}|\mathcal{E}) y_{\hat{\pi}}^j d\hat{\pi} = \int_{\hat{\pi}} P(\hat{\pi}|\mathcal{E}) y_{\hat{\pi}}^j d\hat{\pi} \leq \frac{8\varepsilon}{pA^{S-1}}$. For this value of $j$,

$$P_j(\mathcal{E}'|\mathcal{E}) = 1 - \int_{\hat{\pi}} P_j(\hat{\pi}|\mathcal{E}) y_{\hat{\pi}}^j d\hat{\pi} \geq 1 - \frac{8\varepsilon}{pA^{S-1}} = \frac{1}{2}$$

The overall failure probability in $\mathcal{M}_j$ is thus $P_j(\mathcal{E}') \geq P_j(\mathcal{E})P_j(\mathcal{E}'|\mathcal{E}) \geq \frac{1}{16} > \delta$, which is a contradiction. Therefore, for all $(\varepsilon, \delta)$-PAC algorithm there exists an instance $\mathcal{M}_j$ for which $P_j(\tau > 1/p) \geq 1/8$, hence the expected sample complexity under this instance is larger than $1/8p$.

Let us finally compute the diameter and bias span of MDP $\mathcal{M}_j$. For states $s_1$ and $s_S$, the only (deterministic) policy with finite traveling time from $s_1$ to $s_j$ is the policy which plays action $j_{n-1}$ in state $s_n$ for any $n \in \{2, ...S\}$. As the bottleneck of exploration is the small probability to transition from $s_1$ to $s_2$, the travel time is biggest from $s_1$ to $s_2$. Therefore,

$$D = \min_{\pi:\mathcal{S}\to\mathcal{A}} \mathbb{E}[\min\{t > 0, s_t = s_1\}|s_0 = s_S, \forall t', a_{t'} = \pi(s_{t'})] = \frac{1}{p} + S - 1 = S - 1 + \frac{A^{S-1}}{16\varepsilon}$$

As for the bias span, by fixing $b(s_2) = 0$, the Poisson equations (5) yield for $i \in \{3, ...S\}$ that $b(s_i) = (i-2)g$, and finally $b(s_1) = (S-1)g \leq S$.

$\square$

Figure 5: An SSP-MDP and its average reward transformation

# D  Complements for Section 6

## D.1  Correctness of the stopping rule

*Proof. (Theorem 5)* We first note that $g_{\hat{\pi}_n} = \overline{P}_{\hat{\pi}_n} \overline{r}_{\hat{\pi}_n} = \overline{P}_{\hat{\pi}_n} \left[ \overline{r}_{\hat{\pi}_n} + P_{\hat{\pi}_t} b_n - b_n \right]$ where we use that $\overline{P}_{\hat{\pi}_n} P_{\hat{\pi}_n} = \overline{P}_{\hat{\pi}_n}$.

Since for all $s, a, p_{s,a} b_n \geq L_{s,a}^n(b_n; \delta)$, we have

$$g_{\hat{\pi}_n} \geq \overline{P}_{\hat{\pi}_n} \left[ \overline{r}_{\hat{\pi}_n} + \left( L_{s,\hat{\pi}_n(s)}^n(b_n; \delta) \right)_s - b_n \right] = \overline{P}_{\hat{\pi}_n} \left( \max_a I_{s,a}^{n,\flat}(b_n; \delta) \right)_s$$

by the definition of $\hat{\pi}_n$, and finally $g_{\hat{\pi}_n} \geq \min_s \max_a I_{s,a}^{n,\flat}(b_n; \delta)$.

For $\pi$ a stationary deterministic optimal policy satisfying the optimal Poisson equation (5) (that exists according to Theorem 8.4.3 and 8.4.4 of [18]),

$$g = g_\pi = \overline{P}_\pi \overline{r}_\pi = \overline{P}_\pi \left[ \overline{r}_\pi + P_\pi b_n - b_n \right] \leq \overline{P}_\pi \left[ \overline{r}_\pi + \left( U_{s,\hat{\pi}_n(s)}^n(b_n; \delta) \right)_s - b_n \right]$$

$$\leq \overline{P}_\pi \left( \max_a I_{s,a}^{n,\sharp}(b_n; \delta) \right)_s$$

and finally $g \leq \max_s \max_a I_{s,a}^{n,\sharp}(b_n; \delta)$

Finally, since $g$ is the gain of an optimal policy, $g \geq g_{\hat{\pi}_n}$. $\qquad\qquad\square$

*Proof. (Theorem 6)* By Theorem 5, we have

$$\mathbb{P}\left[ \tau < +\infty, g_{\hat{\pi}} < g^\star - \varepsilon \right]$$

$$\leq \mathbb{P}\left[ \exists n, g^\star - g_{\hat{\pi}_n} > \varepsilon, \max_s \max_a I_{s,a}^{n,\sharp}(b_n; \delta) - \min_s \max_a I_{s,a}^{n,\flat}(b_n; \delta) \leq \varepsilon \right]$$

$$\leq \mathbb{P}\left[ \exists n, \exists s, a, p_{s,a} b_n \notin [L_{s,a}^n(b_n; \delta), U_{s,a}^n(b_n; \delta)] \right]$$

Using further a union bound and the definitions of $U$ and $L$,

$$\mathbb{P}\left[ \tau < +\infty, g_{\hat{\pi}} < g^\star - \varepsilon \right]$$

$$\leq \mathbb{P}\left[ \exists n, \exists s, a, p_{s,a} b_n \geq U_{s,a}^n(b_n; \delta)) \cup \exists n, \exists s, a, p_{s,a} b_n \leq L_{s,a}^n(b_n; \delta)) \right]$$

$$\leq \mathbb{P}\left[ \exists n, \exists s, \exists a, \mathrm{KL}(\hat{p}_{s,a}^n, p_{s,a}) > \frac{x\left(\delta, N_{s,a}^n\right)}{N_{s,a}^n} \right]$$

$$\leq \delta \,,$$

where the last inequality follows from the concentration result given in Lemma 5.

$\qquad\qquad\square$

## D.2  A sample complexity analysis

In this section, we present and analyze an algorithm for the generative model setting using the simplest possible sampling rule, uniform sampling, together with the adaptive stopping rule (6) for $b_n = \hat{b}_n$ such that $\hat{b}_n$ is the optimal bias function in the AR-MDP with transition probabilities given by $(\hat{p}_{s,a}^n)_{s,a}$ (normalized so that the bias in the first state is always 0). The pseudo-code of the algorithm is given in Algorithm 4.

**Remark 1.** *The quantities $U_{s,a}^n$ and $L_{s,a}^n$ in Definition 2 are solutions to complex optimization problems and can be expensive to compute. Several modifications are possible to prevent long run times. First, it is possible not to check for stopping at each time step while preserving correctness. Second, the stopping rule can be relaxed with the following looser confidence intervals which satisfy $U_{s,a}^n(b_n; \delta) \leq \widetilde{U}_{s,a}^n(b_n; \delta)$ and $L_{s,a}^n(b_n; \delta) \geq \widetilde{L}_{s,a}^n(b_n; \delta)$ using Pinsker's inequality:*

$$\widetilde{U}_{s,a}^n(b_n; \delta) = \hat{p}_{s,a}^n b_n + ||b_n||_\infty \sqrt{\frac{2x(\delta, N_{s,a}^n)}{N_{s,a}^n}}, \quad \widetilde{L}_{s,a}^n(b_n; \delta) = \hat{p}_{s,a}^n b_n - ||b_n||_\infty \sqrt{\frac{2x(\delta, N_{s,a}^n)}{N_{s,a}^n}} \,.$$

*Theorem 6 (as well as our sample complexity analysis to follow) still applies for these alternative choices which are easier to compute but yield looser confidence intervals and thus in theory a larger sample complexity.*

**Data:** Accuracy $\varepsilon \in (0,1)$, confidence level $\delta \in (0,1)$
$n = 0$, $b_n = 0$, $I_{s,a}^{n,\sharp}(b_n, \delta) = 1$, $I_{s,a}^{n,\sharp}(b_n, \delta) = 0$;
**while** $\max_s \max_a I_{s,a}^{n,\sharp}(b_n, \delta) - \min_s \max_a I_{s,a}^{n,\flat}(b_n; \delta) > \varepsilon$ **do**
    **for** $(s,a) \in \mathcal{S} \times \mathcal{A}$ **do**
        |   Sample a reward and a next state of $s, a$
    **end**
    $n = n + SA$;
    Compute $\hat{p}_{s,a}^n$ for each $(s,a)$
    Compute $b_n = \hat{b}_n$ the bias of the empirical MDP, with rewards $\overline{r}_{s,a}$ and transitions $\hat{p}_{s,a}^n$;
    Compute $I_{s,a}^{n,\sharp}(b_n, \delta)$ and $I_{s,a}^{n,\flat}(b_n, \delta)$ using $U, L$ from Definition 2 (or the relaxations (7))
**end**
**return** $\hat{\pi}_n = \left( \arg\max_a I_{s,a}^{n,\flat}(b_n; \delta) \right)_s$

**Algorithm 4:** Uniform sampling combined with adaptive stopping

We analyze Algorithm 4 for unichain MDPs, for which we are able to derive a simulation lemma, akin to those existing in the discounted [14] or SSP [21] settings. It relates the gains and the bias in two AR-MDPs that are close enough.

**A simulation lemma for AR-MDPs**    To introduce this result, we need the following definitions.

**Definition 3.** *Let $\mathcal{D}_M$ be the set of policies generating a unichain Markov chain on a weakly communicating MDP $M$. For $M$ an MDP and $\pi \in \mathcal{D}_M$,*

$$
\Pi_\pi = \left( \begin{array}{c|ccc} 1 & 0 & \dots & 0 \\ \hline 1 & & & \\ \dots & & I_{S-1} & \\ 1 & & & \end{array} \right) - \left( \begin{array}{cccc} 0 & P_\pi(s_1, s_2) & \dots & P_\pi(s_1, s_S) \\ \dots & & & \dots \\ 0 & P_\pi(s_S, s_2) & \dots & P_\pi(s_S, s_S) \end{array} \right)
$$

*is the matrix such that the Poisson equations (3) and (4) rewrite as $\Pi_\pi h_\pi = \overline{r}_\pi$ with $\overline{r}$ the average reward vector and $h_\pi = (g_\pi, b_\pi(s_2) - b_\pi(s_1), \dots b_\pi(s_S) - b_\pi(s_1))$. We further define*

$$
\delta_1 = \min_{\pi, g_\pi \neq g^\star} |g_\pi - g^\star|, \quad \Delta_1 = \max_{\pi \in \mathcal{D}_M} ||\Pi_\pi^{-1}|| \quad \text{and} \quad \Delta_2 = \max_{\pi \in \mathcal{D}_M} ||h_\pi||_\infty
$$

*where the norm used is the operator norm associated to the infinity norm, $||A|| = \sup_{||x||_\infty = 1} ||Ax||_\infty$.*

We can show with results from [18] that $\Pi_\pi$ is invertible for any unichain policy $\pi$, hence $\Delta_1$ is well-defined. Indeed, by Theorem A.7 and (A.4), there exist solutions to the Poisson equations. By Theorem 8.2.6, the solution is unique up to an additive vector in the kernel of $(I - P)$, which is of dimension 1 for unichain MDPs as proved in Theorem A.5. Therefore, since in the unichain setting solutions to $\Pi_\pi H_\pi = \overline{r}_\pi$ are exactly solutions for (4), we can conclude.

The following two lemmas can be extracted from the proofs of Lemmas 7 and 8 of [4]. For completeness, we provide a full proof in Appendix D.3.

**Lemma 3** (Simulation lemma). *Let $p' : \mathcal{S} \times \mathcal{A} \to \Sigma_\mathcal{S}$ be the transition probability matrix of a weakly communicating MDP $M'$ with the same reward function $\overline{r}$ as $M$. For a given policy $\pi$ that is unichain on $M$, if for all $(s,a)$ we have $||p'_{s,a} - p_{s,a}||_1 \leq \frac{x}{\Delta_1(\Delta_2 + x)}$ then $||h_\pi - h_\pi^{M'}||_\infty \leq x$.*

**Lemma 4.** *Suppose $M$ **unichain**. Let $p' : \mathcal{S} \times \mathcal{A} \to \Sigma_\mathcal{S}$ be the transition probability matrix of a unichain MDP $M'$ with the same reward function $\overline{r}$ as $M$. If for all $s, a$ we have $||p'_{s,a} - p_{s,a}||_1 \leq \frac{x}{\Delta_1(\Delta_2 + x)}$ where $x \leq \delta_1/2$, then $||b^\star - b^{\star, M'}||_\infty \leq x$.*

It is necessary to suppose $M$ unichain, because without this assumption, there is no guarantee that there exists a policy $\pi$ that is optimal in $M'$ and unichain in $M$.

**Sample complexity bound**    Using these results together with our previous concentration result for the transitions (Lemma 5) permits to prove the following high-probability bound on the sample complexity of Algorithm 4, which features the quantity $\Gamma_\mathcal{M} := \Delta_1(\Delta_2 + \delta_1/2)$.

**Theorem 9.** *For $\varepsilon \leq \delta_1$, with probability larger than $1 - \delta$, the sample complexity of Algorithm 4 on a unichain MDP satisfies*

$$\tau = \widetilde{\mathcal{O}}\left(\frac{SA((H+1) \vee \Gamma_{\mathcal{M}})^2}{\varepsilon^2}\left(\log\left(\frac{1}{\delta}\right) + S\right)\right) .$$

Just like DFE, Algorithm 4 is an $(\varepsilon, \delta)$-PAC algorithm using a generative model that does not require any prior knowledge on the MDP, as its stopping rule is fully data-dependent. Their sample complexity in the regime of small $\varepsilon$ and small $\delta$ both scale in $(SAc(\mathcal{M})/\varepsilon^2)\log(1/\delta)$ but for different complexity quantities: $c_1(\mathcal{M}) = D$ for Algorithm 1 while $c_2(\mathcal{M}) = ((H+1) \vee \Gamma_{\mathcal{M}})^2$ for Algorithm 4. We know from the lower bound that the latter has to be larger for some MDPs, but so far we did not manage to quantify their difference (even if we suspect that $c_2$ can be much larger than $c_1$). Besides this, we hope that the sample complexity of Algorithm 4 can be significantly reduced by the use of smarter (online) sampling rules.

### D.3 Simulation lemmas

*Proof. (Lemma 3)* First, let us fix any policy $\pi$ and $y < \frac{1}{||\Pi_\pi^{-1}||}$ such that $\max_{s,a}||p_{s,a} - p'_{s,a}||_1 \leq y$. We thus have $\Pi_\pi h_\pi = \Pi'_\pi h_\pi^{M'}$. $\Pi_\pi - \Pi'_\pi = P'_\pi - P_\pi$, and therefore $||\Pi_\pi - \Pi'_\pi|| < y$.

$$h_\pi(\Pi_\pi - \Pi'_\pi) = \Pi'_\pi(h_\pi^{M'} - h_\pi)$$
$$||\Pi'^{-1}_\pi|| \cdot ||h_\pi|| \cdot y > ||h_\pi^{M'} - h_\pi||_\infty$$

Moreover,

$$||\Pi'^{-1}_\pi|| \leq ||\Pi_\pi^{-1}|| \cdot ||\Pi_\pi \Pi'^{-1}_\pi|| - ||\Pi_\pi^{-1}|| + ||\Pi_\pi^{-1}||$$
$$||\Pi'^{-1}_\pi|| \leq ||\Pi_\pi^{-1}|| \cdot ||\Pi_\pi \Pi'^{-1}_\pi - I|| + ||\Pi_\pi^{-1}||$$
$$||\Pi'^{-1}_\pi|| \leq ||\Pi_\pi^{-1}|| \cdot ||\Pi'^{-1}_\pi|| \cdot y + ||\Pi_\pi^{-1}||$$
$$||\Pi'^{-1}_\pi|| \leq \frac{||\Pi_\pi^{-1}||}{1 - ||\Pi_\pi^{-1}|| \cdot y}$$

since $||\Pi_\pi^{-1}|| \cdot y < 1$.

And therefore,

$$||h_\pi - h_\pi^{M'}|| < \frac{||\Pi_\pi^{-1}|| \cdot ||h_\pi|| \cdot y}{1 - ||\Pi_\pi^{-1}|| \cdot y} \leq \frac{\Delta_1 \Delta_2 y}{1 - \Delta_1 y}$$

Therefore, for any $x$, taking $y = \frac{x}{\Delta_1 \Delta_2 + x \Delta_1}$ gives the result. □

*Proof. (Lemma 4)* Fix $x < \delta_1/2$, and let $\pi$ be an optimal policy in $M'$. For any policy $\pi'$, using Lemma 3 successively with policy $\pi$ and policy $\pi'$ yields $g_\pi > g'_\pi - x \geq g'_{\pi'} - x > g_{\pi'} - 2x$ and finally $g_\pi > g_{\pi'} - \delta_1$, which implies by definition of $\delta_1$ that $g_\pi \geq g_{\pi'}$, and therefore $\pi$ is optimal in $M$. Hence applying Lemma 3 to $\pi$ yields the result. □

### D.4 Proof of Theorem 9

Fix a unichain MDP. We prove in Theorem 10 below an explicit upper bound on the sample complexity of Algorithm 4. For $\varepsilon \leq \delta_1$, we deduce from it that

$$\tau = \widetilde{\mathcal{O}}\left(\frac{SA((H+1) \vee \Gamma_{\mathcal{M}})^2}{\varepsilon^2}\left(\log\left(\frac{1}{\delta}\right) + S\right)\right) ,$$

as claimed in Theorem 9.

**Theorem 10.** *With probability larger than $1 - \delta$, the sample complexity of Algorithm 4 is smaller than*

$$SA(S-1)\frac{y}{C}\left(1 + \frac{2C}{y}\log\left(\frac{y+2}{C}\right)\right)$$

*where $y = 1 + \frac{\ln SA/\delta}{S-1}$ and $C = \frac{1}{288}\frac{\min(\delta_1,\varepsilon)^2}{\max(H+1,\Gamma_{\mathcal{M}})^2}$.*

**Proof of Theorem 10**   We write $N_t = \lfloor \frac{t}{SA} \rfloor$ to denote the number of visits in each state action pair at time step $t$. We denote by $b$ the optimal bias function (normalized to be equal to zero in the first state). We set $\xi \in (0, \frac{1}{6})$ to be determined later and define the three events

$$\mathcal{E} = \left\{ \forall t, \forall s, a, ||\hat{p}^t_{s,a} - p_{s,a}||_1 \leq \sqrt{\frac{2x(\delta, N_t)}{N_t}} \right\}$$

$$\mathcal{E}_1(\xi, t) = \left\{ ||\hat{b}_t - b||_\infty < \xi \varepsilon \right\}$$

$$\mathcal{E}_2(\xi, t) = \left\{ \forall s, a, ||\hat{b}_t||_\infty \sqrt{2 \frac{x(\delta, N_t)}{N_t}} < \min \left( \left( \frac{1}{4} - \frac{3}{2}\xi \right) \varepsilon, ||\hat{b}_t||_\infty \right) \right\}$$

Using Lemma 5, we know that $\mathbb{P}(\mathcal{E}) \geq 1 - \delta$.

**We first prove that, if $\mathcal{E} \cap \mathcal{E}_1(\xi, t) \cap \mathcal{E}_2(\xi, t)$ is satisfied for a certain time step $t$, then the stopping condition (6) is satisfied at this time step.** To this end, let us assume that $\mathcal{E} \cap \mathcal{E}_1(\xi, t) \cap \mathcal{E}_2(\xi, t)$ holds for a fixed $t$ and $\xi$. We recall the definition of the confidence bounds $U$ and $L$ from Definition 2 and their relaxations $\widetilde{U}$ and $\widetilde{L}$ defined in (7). For all $(s, a)$, we have

$$\overline{r}_{s,a} + U^t_{s,a}(\hat{b}_t; \delta) - \hat{b}_t(s) \leq \overline{r}_{s,a} + \widetilde{U}^t_{s,a}(\hat{b}_t; \delta) - \hat{b}_t(s)$$

$$= \overline{r}_{s,a} + \hat{p}^t_{s,a}\hat{b}_t + \sqrt{\frac{2x(\delta, N^t_{s,a})}{N^t_{s,a}}}||\hat{b}_t||_\infty - \hat{b}_t(s)$$

$$\leq \overline{r}_{s,a} + p_{s,a}b + (\hat{p}^t_{s,a} - p_{s,a})b + \hat{p}^t_{s,a}(\hat{b}_t - b)$$

$$+ \sqrt{\frac{2x(\delta, N^t_{s,a})}{N^t_{s,a}}}||\hat{b}_t||_\infty - \hat{b}_t(s) + b(s) - b(s)$$

$$\leq \overline{r}_{s,a} + p_{s,a}b - b(s) + ||b - \hat{b}_t + \hat{b}_t||_\infty \sqrt{\frac{2x(\delta, N^t_{s,a})}{N^t_{s,a}}}$$

$$+ ||\hat{b}_t - b||_\infty + \sqrt{\frac{2x(\delta, N^t_{s,a})}{N^t_{s,a}}}||\hat{b}_t||_\infty + ||\hat{b}_t - b||_\infty$$

$$\leq \overline{r}_{s,a} + p_{s,a}b - b(s) + \xi\varepsilon + \left( \frac{1}{4} - \frac{3}{2}\xi \right)\varepsilon + \xi\varepsilon + \left( \frac{1}{4} - \frac{3}{2}\xi \right)\varepsilon + \xi\varepsilon$$

$$\leq \overline{r}_{s,a} + p_{s,a}b - b(s) + \frac{1}{2}\varepsilon$$

Similarly, we can prove that

$$\overline{r}_{s,a} + L^t_{s,a} - \hat{b}_t(s) \leq \overline{r}_{s,a} + p_{s,a}b - b(s) - \frac{1}{2}\varepsilon.$$

Finally, since $b$ is a solution to the optimal Poisson equation (5), we know that

$$\max_s \max_a \left( \overline{r}_{s,a} + p_{s,a}b - b(s) \right) - \min_s \max_a \left( \overline{r}_{s,a} + p_{s,a}b - b(s) \right) = 0$$

It follows that

$$\max_s \max_a \left( \overline{r}_{s,a} + U^t_{s,a}(\hat{b}_t; \delta) - \hat{b}_t(s) \right) - \min_s \max_a \left( \overline{r}_{s,a} + L^t_{s,a}(\hat{b}_t; \delta) - \hat{b}_t(s) \right) \leq \varepsilon$$

and the stopping condition is met.

**Then, we establish a sufficient condition on $\xi$ and $t$ to have $\mathcal{E} \subseteq \mathcal{E}_1(\xi, t)$.**

Assume that the event $\mathcal{E}$ holds. Let $\xi' = \min(\xi\varepsilon, \delta_1/2)$. From Lemma 4, if for all $s, a$ we have $||\hat{p}_t(s, a) - p(s, a)||_1 \leq \frac{\xi'}{\Gamma_\mathcal{M}}$, then $||b - \hat{b}_t||_\infty \leq \xi'$, which implies $\mathcal{E}_1(\xi, t)$. Hence, on $\mathcal{E}$, a sufficient condition for $\mathcal{E}_1(\xi, t)$ to hold is

$$\sqrt{\frac{2x(\delta, N_t)}{N_t}} < \frac{\xi'}{\Gamma_\mathcal{M}}.$$

Introducing $c(\xi) = \frac{1}{2}\left(\frac{\xi'}{\Gamma_\mathcal{M}}\right)^2$, this is equivalent to

$$\log(SA/\delta) + (S-1)\left(1 + \log(1 + N_t/(S-1))\right) \leq c(\xi)N_t$$

and finally to

$$c(\xi)\frac{N_t}{S-1} - \log\left(1 + \frac{N_t}{S-1}\right) \geq 1 + \frac{\log(SA/\delta)}{S-1} \ . \tag{10}$$

**Next, we establish a sufficient condition on $\xi$ and $t$ to have $\mathcal{E}_1(\xi, t) \subseteq \mathcal{E}_2(\xi, t)$.** We first remark that when $\mathcal{E}_1(\xi, t)$ holds

$$
\|\hat{b}_t\|_\infty \sqrt{\frac{2x(\delta, N_t)}{N_t}} \quad \leq \quad \left(\|\hat{b}_t - b\|_\infty + \|b\|_\infty\right)\sqrt{\frac{2x(\delta, N_t)}{N_t}}
$$

$$
\leq \quad (\xi\varepsilon + H)\sqrt{\frac{2x(\delta, N_t)}{N_t}}
$$

Therefore, a sufficient condition for $\mathcal{E}_2(\xi, t)$ to hold is

$$\sqrt{\frac{2x(\delta, N_t)}{N_t}} < \min\left(\frac{1 - 6\xi}{4(H + \xi\varepsilon)}\varepsilon, 1\right)$$

Introducing $c'(\xi) = \min\left(\frac{1}{32}\left(\frac{1-6\xi}{H+\xi\varepsilon}\right)^2\varepsilon^2, \frac{1}{2}\right)$, this is equivalent to

$$\log(SA/\delta)/(S-1) + 1 + \log(1 + N_t/(S-1)) < c'(\xi)N_t/(S-1)$$

and finally to

$$c'(\xi)\frac{N_t}{S-1} - \log\left(1 + \frac{N_t}{S-1}\right) > 1 + \frac{\log(SA/\delta)}{S-1} \ . \tag{11}$$

**Finally, we put things together.** The conditions (10) and (11) are of similar form. Defining $C(\xi) = \min(c(\xi), c'(\xi))$, for any $\xi \in (0, 1/6)$ the condition

$$C(\xi)\frac{N_t}{S-1} - \log\left(1 + \frac{N_t}{S-1}\right) > 1 + \frac{\log(SA/\delta)}{S-1} \tag{12}$$

is a sufficient condition on $t$ to have $\mathcal{E} \subseteq \mathcal{E} \cap \mathcal{E}_1(\xi, t) \cap \mathcal{E}_2(\xi, t)$. If follows that for $t$ satisfying (12), the algorithm has stopped before time $t$, with probability larger than $1 - \delta$.

We observe that the larger $C(\xi)$, the less stringent the condition is on $t$, but finding the value of $\xi \in (0, 1/6)$ that maximizes $C(\xi)$ leads to tedious calculations. Instead we go for finding a lower bound on $C(1/12)$, denoted by $C$, which also provides a valid sufficient condition on $t$:

$$C\frac{N_t}{S-1} - \log\left(1 + \frac{N_t}{S-1}\right) > 1 + \frac{\log(SA/\delta)}{S-1}.$$

Using Lemma 6, letting $y = 1 + \frac{\log(SA/\delta)}{S-1}$, this condition is satisfied for

$$\frac{N_t}{S-1} \geq \frac{y}{C}\left(1 + \frac{2C}{y}\log\left(\frac{y+2}{C}\right)\right) \ .$$

To conclude the proof, we explicit the value of $C$. We have

$$
\begin{aligned}
C(1/12) &= \min\left[\frac{1}{2}, \frac{1}{32}\frac{(1/2)^2}{(H + \varepsilon/12)^2}\varepsilon^2, \frac{1}{2}\frac{\min[(\varepsilon/12)^2, (\delta_1/2)^2]}{\Gamma_\mathcal{M}^2}\right] \\
&= \min\left[\frac{1}{2}, \frac{1}{128}\frac{\varepsilon^2}{(H + \varepsilon/12)^2}, \frac{\varepsilon^2}{288\Gamma_\mathcal{M}^2}, \frac{\delta_1^2}{8\Gamma_\mathcal{M}^2}\right] \\
&\geq \min\left[\frac{1}{2}, \frac{1}{128}\frac{\varepsilon^2}{(H + 1)^2}, \frac{\varepsilon^2}{288\Gamma_\mathcal{M}^2}, \frac{\delta_1^2}{8\Gamma_\mathcal{M}^2}\right] \\
&= \min\left[\frac{1}{128}\frac{\varepsilon^2}{(H + 1)^2}, \frac{\varepsilon^2}{288\Gamma_\mathcal{M}^2}, \frac{\delta_1^2}{8\Gamma_\mathcal{M}^2}\right],
\end{aligned}
$$

where we used twice that $\varepsilon \leq 1$. Hence a valid lower bound on $C(1/12)$ is

$$C = \frac{1}{288} \frac{\min(\delta_1, \varepsilon)^2}{\max(H+1, \Gamma_{\mathcal{M}})^2}.$$

□

## E  Auxillary results

In this section, we restate some useful result from the literature. The first is a time uniform concentration result on the transition probabilities that can be obtained as a consequence of Lemma 9 from [1].

**Lemma 5.** *For all $\delta \in (0,1)$, with $x(\delta, y) = \log(SA/\delta) + (S-1)\log(e(1 + y/(S-1)))$,*

$$\mathbb{P}\left[\exists n \in \mathbb{N} : N_{s,a}^n \mathrm{KL}(\hat{p}_{s,a}^n, p_{s,a}) > x(\delta, N_{s,a}^n)\right] \leq \frac{\delta}{SA}$$

*Furthermore, letting $B(n, \delta) := \sqrt{\frac{2x(\delta, n)}{n}}$, the event*

$$\mathcal{E} = \left(\forall s, a, \forall n \geq 1, \|\hat{p}_{s,a}^n - p_{s,a}\|_1 \leq B(N_{s,a}^n, \delta)\right)$$

*holds with probability $1 - \delta$.*

*Proof.* With $\hat{p}_{s,a}(m)$ the empirical estimate of $p_{s,a}$ once $m$ samples have been collected in $s, a$, arbitrarily setting $\hat{p}_{s,a}(0)$ as the constant vector $1/S$,

$$\begin{aligned}
\mathbb{P}\big[\exists n \in \mathbb{N} : &N_{s,a}^n \mathrm{KL}(\hat{p}_{s,a}^n, p_{s,a}) > x(\delta, N_{s,a}^n, S)\big] \\
&\leq \mathbb{P}\left[\exists n, m \in \mathbb{N} : N_{s,a}^n = m, N_{s,a}^n \mathrm{KL}(\hat{p}_{s,a}^n, p_{s,a}) > x(\delta, N_{s,a}^n, S)\right] \\
&\leq \mathbb{P}\left[\exists m \in \mathbb{N} : m\mathrm{KL}(\hat{p}_{s,a}(m), p_{s,a}) > x(\delta, m, S)\right] \\
&\leq \frac{\delta}{SA}
\end{aligned}$$

by Lemma 9 from [1]. The second claim stems from a union bound over $S, A$ and Pinsker's inequality applied to the first claim.

□

The second result is an inversion lemma, which allows to get explicit sample complexity bounds.

**Lemma 6** (Lemma 15 from [13]). *Let $n \geq 1$ and $a, b, c, d > 0$.*

*If $n\Delta^2 \leq a + b\log(c + dn)$ then*

$$n \leq \frac{1}{\Delta^2}\left[a + b\log\left(c + \frac{d}{\Delta^4}(a + b(\sqrt{c} + \sqrt{d}))^2\right)\right]$$

