# OpenReview forum: "Finding good policies in average-reward Markov Decision Processes without prior knowledge"
_NeurIPS.cc/2024/Conference — NeurIPS 2024 poster_

### Official Review · Reviewer_FXz4 · 2024-07-11

**Soundness:** 3
**Presentation:** 4
**Contribution:** 3
**Rating:** 6
**Confidence:** 4

**Summary:**

The goal of the paper is to learn near-optimal policies in average reward MDPs without prior knowledge of complexity parameters, contrasting extensive prior work which requires knowledge of the values complexity parameters. They rule out the possibility of easily removing this knowledge requirement through estimation of the optimal bias span, but they show that the diameter can be estimated, and use it to give an algorithm in the generative model setting. In the online setting, they again show a negative result for achieving a span-based complexity, but build on the ideas from the generative setting to give a diameter-based complexity bound. They also propose a stopping rule and give some preliminary analysis of its performance.

**Strengths:**

The problem of learning near-optimal policies in average reward MDPs without knowledge of complexity parameters is a fundamental and important problem, and this paper makes a number of small but important steps towards either achieving this goal or demonstrating its difficulty. I therefore am confident that this work will prove to be significant, in the sense that it will lead to future work which build off of its attempts. I think the presentation is mostly very clear and does a good job motivating the different settings and ideas.

**Weaknesses:**

Algorithm 1 is not very novel, since it combines two ingredients from prior work: an optimal algorithm when knowledge of the optimal bias span is known, and a diameter estimation algorithm.

The value of including section 6.2 (the value iteration-inspired stopping rule) is not clear to me, since the algorithm is only given a proof-of-concept analysis for the generative model setting (not the motivating online setting), and furthermore it involves seemingly foreign complexity parameters and does not improve upon the other introduced methods.

**Questions:**

Line 110: In weakly communicating MDPs there is guaranteed to be a policy with optimal average reward which is unichain, however this policy might not be any of the higher-order notions of optimality (ex. bias optimal or Blackwell optimal). Also/therefore note that multiple policies with optimal gain may not have the same bias. Thus I think a more careful definition of the optimal bias span is needed here. Also I don't understand why the optimal policy is assumed to be unique; this is clearly not true in many situations (especially in average reward settings, there may be multiple ways to reach the same optimal recurrent class) and would greatly limit the applicability of the results. It doesn't seem like this assumption is needed for anything except potentially convenience in definitions, so I think it should be removed.

This is very nitpicky, but I find it a bit weird that some bounds in Table 1 do not have a listed dependence on $\log(1/\delta)$ (ex.  [27], which apparently does actually depend on $\log(1/\delta)$). Anyways, why include the dependence on $\log(1/\delta)$ for the other results if the $\tilde{O}$ notation is meant to hide log factors?

Theorem 1 (impossibility of estimating $H$) uses unknown rewards, but the previously described setting assumes known mean rewards. Is it possible to get this result to work with known rewards? (One can try replacing the unknown reward state with a sub-MDP with unknown dynamics, but the part which is unclear to me is how adding this sub-MDP would affect the bias span)

The formatting for the display for Algorithm 1 should be fixed

**Limitations:**

There are no significant limitations which require addressing

---

> ### Author Rebuttal · Authors · 2024-08-06
>
> We are thankful to the reviewer for their very constructive feedback on our paper.
>
> While the algorithms we present are combinations of previously known methods, the main point of interest of this paper is that $H$ is not the right measure of complexity due to the impossibility of estimating it and the fact the sample complexity does not scale well with it in the online setting. %The solutions given are indeed proofs of concept.
> We show that combining prior work is near optimal with regards to the more meaningful complexity measure $D$. While the complexity analysis in 6.2 is only valid in the generative model, the correctness is proved also in the online setting, and thus provides a new way of constructing online best policy identification algorithms.
>
> The assumption of unique optimal policy is indeed merely useful for convenience of definitions. We can remove this assumption by redefining $H$ to be the maximum bias span over all gain-optimal policies, as we do not try to reach bias-optimality.
>
> The notation in Table 1 is meant to hide log factors in $\log(1/\delta)$, not those in $\delta$ (as $\log(1/\delta)$ is the ``right'' dependence in $\delta$). We will clarify this in the paper. The bound in [27] depends on $\log(1/\delta)$ and that dependence is missing in the table: this will be fixed. The other entry that currently does not have a $\log(1/\delta)$ factor is the lower bound of [12], which indeed does not depend on $\delta$.
>
> For Theorem 1, we can change the example to obtain a MDP with known rewards, and shift the difficulty to estimating transitions. We can replace state 2 by three states, (2, 2', 2''), where 2 is connected to 1 and 3 as in Figure 1, but we remove the action with reward R.
> Instead we add an action with reward 1/2 that transitions to 2' with probability $R$ and to 2'' with probability $1-R$. In 2' there is only one action which goes to 2 with probability 1 and reward 1. In 2'' there is also one action which goes to 2 with probability 1 and reward 0. Choosing $p$ to be $(1+\varepsilon)/2\Delta$ and $R=1/2 \pm\varepsilon$ yields the result through the same reasoning as in Theorem 1.

---

> > ### Comment · Reviewer_FXz4 · 2024-08-10
> >
> > Thanks for the response. I am not sure if the proposed solution for redefining $H$ is satisfactory, because the paper concerns the estimation of $H$ and this redefinition seems like it may affect the difficulty of this task. Again, I think assuming a unique optimal policy is a very bad assumption, as this is basically assuming that gain optimal $\implies$ bias optimal. Overall, I will maintain my score.

---

> > > ### Comment · Reviewer_XRXp · 2024-08-13
> > >
> > > It is not clear to me why redefining H is unsatisfactory. From the lower bounds perspective, defining H as the maximum of bias spans of all gain optimal policies just makes the lower bound stronger. The paper makes no claims of using H in upper bounds, so I am truly puzzled why this redefinition is unsatisfactory.

---

> > > > ### Comment · Reviewer_FXz4 · 2024-08-13
> > > >
> > > > One result in the paper concerns estimating $H$. If the definition of $H$ is changed to a nonstandard one, then I think this result has less meaning or more unclear meaning. Also note that it is possible to construct example MDPs where some gain optimal policy has bias span which is larger than $D$ (say there is an absorbing state with reward 1, then a gain optimal policy only needs to get there eventually but can be very suboptimal and take a very long time to get there). Thus if the new definition of $H$ is used, we do not have the relationship $H \leq D$. This is further evidence that the proposed definition is not satisfactory.

---

> > > > > ### Author Response · Authors · 2024-08-14
> > > > >
> > > > > Sorry for the confusion. We use the standard definition for the bias span H, that was shown in the Tewari and Bartlett paper of 2009, REGAL: A Regularization based Algorithm for Reinforcement Learning in Weakly Communicating MDPs,  to be smaller than the diameter. We indeed do not require the unique optimal policy assumption, and can remove this assumption. In our paper, we focus solely on gain-optimal policies, not bias-optimal policies. Thank you for your constructive comments.

---

### Official Review · Reviewer_en8V · 2024-07-11

**Soundness:** 3
**Presentation:** 2
**Contribution:** 2
**Rating:** 4
**Confidence:** 4

**Summary:**

This paper studies the problem of learning a good policy in averaged-reward MDP with finite diameter. Given previous work on the problem assuming knowledge on the optimal bias span $H$, the authors try to remove the prior knowledge and propose diameter-dependent sample complexity without any prior knowledge. In the meantime, the authors present several observations about the hardness to reach better sample complexity bounds.

**Strengths:**

I appreciate the efforts to study the sample complexity bounds without prior knowledge.  However, the current result seems incremental and insufficient for an acceptance in my opinion.

**Weaknesses:**

The major concern is due to limited contribution. Two possible directions to improve this work: (1) Is $H$-dependent sample complexity reachable without prior-knowledge (even allowing worse dependence on $1/\epsilon$)? It might be hard to estimate $H$ precisely according to your observations, but that does not means a lower bound. (2) It seems the exact $D$ factor in the online case could be removed by some efficient reward-free exploration algorithms. One could assume an approximate transition model to help conduct reward-free exploration, where the target is to solve the following problem $\min_{\pi_b}\max_{\pi}\sum_{s,a}\frac{d_{\pi,T}(s,a)}{d_{\pi_b,T}(s,a)}$  ($d_{\pi,T}(s,a)$ is the occupancy distribution following $\pi$ in $T$ steps). I think the following papers might help solve the problem efficiently.

Minimax-optimal reward-agnostic exploration in reinforcement learning, Li et.al., 2024;

Horizon-Free Reinforcement Learning in Polynomial Time: the Power of Stationary Policies, Zhang et. al., 2022

**Questions:**

Please find my questions in the comments above.

**Limitations:**

Yes.

---

> ### Author Rebuttal · Authors · 2024-08-06
>
> We appreciate the reviewer's insightful feedback on our paper.
>
> Our main contribution in this paper is that H is not the right complexity measure for best policy identification in average reward MDPs. Once that point is made, we then turn to other measures and provide an algorithm that shows that a bound depending on D instead can be attained without prior knowledge.
>
> It is true that we have not shown that there cannot be any algorithms scaling in $H$ in the generative model. However, we have seen that there are none in the current literature. Since the original lower bound of [22] scales in $D$, $H$ is impossible to estimate, and the in the online setting an algorithm can reach $D$ but not $H$, we postulate that bounds scaling in $H$ are unattainable. However, what is certain is that the current state of the art algorithms cannot actually scale in $H$, making our algorithm scaling in $D$ meaningful.
>
> We thank the reviewer for the references to reward-free exploration and will investigate how they could be used for best policy identification.

---

> > ### Comment · Reviewer_en8V · 2024-08-10
> >
> > Thanks for the response. Current I  would like to keep the score. I will adjust the score after discussion with AC and other reviewers.

---

### Official Review · Reviewer_NtfV · 2024-07-12

**Soundness:** 4
**Presentation:** 3
**Contribution:** 2
**Rating:** 4
**Confidence:** 3

**Summary:**

This paper addresses average-reward Markov Decision Processes (MDPs). In the context of the generative model, existing literature presents an $\epsilon$-optimal policy with a sample complexity of $O(SAD/\epsilon^2)$. However, this approach requires prior knowledge of an upper bound on the optimal bias span $H$. This paper initially demonstrates that accurately estimating $H$ can have arbitrarily large complexity. Subsequently, it introduces the Diameter Free Exploration (DFE) algorithm for communicating MDPs, which operates without any prior knowledge about the MDP and achieves near-optimal sample complexity for small $\epsilon$. In the online setting, the authors establish a lower bound suggesting that achieving a sample complexity polynomial in $H$ is infeasible. Furthermore, they propose an online algorithm that attains a sample complexity of $O(SAD^2/\epsilon^2)$. They also propose a data-dependent stopping rule that they believe could reduce sample complexity in the online setting.

**Strengths:**

This paper presents a complete procedure, DFE, that achieves near-optimal sample complexity for small $\epsilon$ without any prior knowledge of $H$ for average-reward MDP. This approach fills a gap in the existing literature.

The finding that accurately estimating $H$ can have arbitrarily large complexity is new and interesting.

This paper also presents a new finding that achieving a sample complexity polynomial in $H$ in the online setting is infeasible, setting theoretical boundaries for future research.

The paper is technically sound, demonstrating rigor in the development and justification of its claims. The presentation quality of this paper is good.

The authors did a great literature review on average-reward MDPs.

**Weaknesses:**

The main concern with this paper is the limited novelty and contribution of the proposed solutions.

The paper's claim regarding the necessity of knowing $H$ in the algorithm from [27] is somewhat misleading. The referenced paper explicitly mentions that only an upper bound for $H$ is required, not precise knowledge of $H$.

The technique of using an upper bound of $D$ to estimate $H$, as presented in this work, is not original. Reference [25] previously introduced this idea, diminishing the perceived innovation of the current paper’s methodology. Additionally, Algorithm 2, designed to estimate $D$, closely resembles Algorithm 4 in [21], suggesting a lack of substantial differentiation in their algorithm design.

The main algorithm, Diameter Free Exploration (DFE), seems to be a straightforward combination of algorithms in [21] and [27]. The primary theoretical contribution, Theorem 2, appears to be direct. The theorem’s proof does not seem to require substantial intellectual effort, suggesting that similar results could be easily derived by others familiar with the cited works.

In the online setting, the situation is similar. The online-DFE primarily integrates existing algorithms with minimal modification, and the theoretical insights it offers do not extend far beyond established results. This recombination of known techniques without substantial new insights significantly diminishes the paper's novelty and impact.

Theorem 3 in this paper is also very similar to Theorem 9 in [5].

While the authors propose a data-dependent stopping rule that they believe could reduce sample complexity in the online setting, they defer its exploration to future work. While postponing this analysis is understandable, it cannot be recognized as a substantial contribution within the current paper.

**Questions:**

Regarding the development of DFE and online-DFE, and Theorems 2 and 4: Is it accurate to say that anyone familiar with [20], [21], and [27] could readily replicate these results? Specifically, were there unique challenges or complexities that are not immediately apparent but critical to the contributions of this paper?

Could the authors clarify which algorithms and theorems they consider to be the main technical contributions of this paper?

**Limitations:**

The authors adequately addressed the limitations in the conclusion.

---

> ### Author Rebuttal · Authors · 2024-08-06
>
> We thank the reviewer for their detailed feedback on our paper.
>
> Our main point in this paper is that H is not the right complexity measure for best policy identification in average reward MDPs. Once that point is made, we then turn to other measures and provide an algorithm that shows that a bound depending on D instead can be attained without prior knowledge.
>
> In [27] and other previous papers, the lower bound considered uses H as the main complexity measure, and the algorithm of [27] is said to have a sample complexity bound that also depends on H, provided H is given to the algorithm.
> As the reviewer points out, the algorithm also can take an upper bound of H and have a sample complexity that depends on the upper bound (and we use this in our paper).
> Nonetheless, the focus in that previous paper and others is on H: H is presented as the right complexity measure, and the need to know an upper bound on it in the algorithm is minimized (H is assumed known "for simplicity").
>
> We argue that having an upper bound on H that actually reflects H (and not something else like D) is not feasible, and that the focus on H in the bounds in the literature is perhaps misplaced. In the generative setting, we prove that we can't first estimate H and then use it in an algorithm that takes it as a parameter (Theorem 1). In the online setting, we prove a lower bound that shows that no algorithm can have an upper bound polynomial in H and not D (Theorem 3).
>
> Then we demonstrate that a weaker complexity measure, D, is attainable. We agree that our algorithms are a combination of previous ones. The novelty in our work is the idea that since H seems to not be attainable, obtaining algorithms that depend on D is meaningful. Given that it's achievable by a combination of previous algorithmic elements, we did not reinvent a new way of doing it (the combination of those methods to obtain an algorithm scaling with D had however never been described).

---

> > ### Comment · Reviewer_XRXp · 2024-08-13
> >
> > The review of reviewer NtfV has the following structure: Acknowledging that the paper makes interesting, meaningful and original contributions in a well written paper. Next, the reviewer complains about that the paper builds heavily on existing work, and the results are achieved with, what they think, is essentially too little effort, resulting in a verdict to reject the paper (rating of 4). As a fellow reviewer, I find this unreasonable. I think we should cherish meaningful, interesting, original findings, even if the results are using tools and techniques that are well established. There is much to like about a paper besides whether it introduces entirely new ideas. And I think the lower bounds in this paper do have some interesting new twists to them, which is overlooked by the reviewer. I hope that reviewer NtfV will change their harsh rating in light of this: Our field does not need to be adversarial. We all build on previous results in smaller or bigger ways.

---

### Official Review · Reviewer_XRXp · 2024-07-15

**Soundness:** 4
**Presentation:** 4
**Contribution:** 4
**Rating:** 8
**Confidence:** 3

**Summary:**

The problem of identifying near optimal policies with high probability, either in the generative, or in the online setting, is considered when the state-action space is finite, and the criteria to compare policies is how much reward they collect on the average in the long term ("PAC setting"). Algorithms are compared based on their sample complexity: the number of interactions they need before they return with a near-optimal policy. The main question is whether knowledge of the span of the optimal value function allows for a reduced sample complexity.
Some partial answers are obtained: For the generative setting, it is shown that estimating the span itself is not tractable. Next, the more moderate goal of designing an algorithm that adapts to the diameter is achieved.
For the online setting it is shown that with or without the knowledge of H, the problem is intractable. Finally, sound algorithms are designed that control sample complexity in terms of the (possibly unknown) diameter.
The paper also explains the difficulty of reducing the PAC problem to cumulative regret minimization.

**Strengths:**

Novel results, new ideas, especially with the lower bounds.

**Weaknesses:**

It was known that one can estimate the diameter; hence algorithms that adapt to the diameter are not that surprising. The PAC setting is a bit artificial. Results for the PAC setting are more interesting if they mimic the results of the other settings (fixed, known or unknown budget, simple regret, or cumulative regret); and the results in this paper just underline how unnatural the PAC setting is (the algorithm needs to know how well it does; this is nice to have, but less essential than doing well.)

Since there is no box to present my summary opinion, I note it here that the above does not mean that there is no reason to study the PAC setting (ie I consider the above a really minor point). In fact, I find the question of whether there is a real difference between the PAC and the other settings an interesting and important question. Overall, I think the paper makes important and interesting contributions.

**Questions:**

n.a.

**Limitations:**

n.a.

---

> ### Author Rebuttal · Authors · 2024-08-06
>
> We thank the reviewer for their thoughtful feedback on our paper.
>
> It is true that we do not relate the PAC setting to most other settings, but we do point out that the results in the cumulative regret setting are unapplicable here, which is quite different from the finite horizon or discounted models. An interesting new preprint, Achieving Tractable Minimax Optimal Regret in Average Reward MDPs by Boone et al., 2024, even provides a regret-minimizing algorithm that scales in $H$ without requiring prior knowledge of it - something we argue does not seem attainable in best policy identification. Looking at what happens when attempting to minimize for example simple regret in average-reward would be interesting future work.

---

> > ### Comment · Reviewer_XRXp · 2024-08-10
> > **Rebuttal read**
> >
> > I confirm I read the rebuttal and the other reviews. I still maintain this is a fine paper investigating a delicate issue in a thoughtful manner.

---

### Decision · Program_Chairs · 2024-09-25

**Decision:**

Accept (poster)

**Comment:**

Based on the reviews and extensive discussions, this paper addresses an important problem in average-reward Markov Decision Processes (MDPs), particularly in identifying near-optimal policies without prior knowledge of complexity parameters. While some reviewers raised concerns about the novelty and the reliance on existing methods, others highlighted the meaningful contributions and the thoughtful investigation into the complexities of the PAC setting in average-reward MDPs. Despite the debates around the necessity of assuming a unique optimal policy and the potential for refining the exposition, the consensus leans toward the paper being a solid contribution that will likely influence future work in the area. Therefore, I recommend accepting this paper, acknowledging its technical soundness and the significance of its contributions, even if they build on established techniques.